# *L*-Arginine and asymmetric dimethylarginine (ADMA) transport across the mouse blood-brain and blood-CSF barriers: Evidence of saturable transport at both interfaces and CNS to blood efflux

**Mehmet Fidanboylu[1,2], Sarah Ann Thomas[1,2]***

**1** Pharmaceutical Sciences Research Division, King's College London, London, United Kingdom, **2** Institute of Pharmaceutical Science, King's College London, London, United Kingdom

* sarah.thomas@kcl.ac.uk

## Abstract

*L*-Arginine is the physiological substrate for the nitric oxide synthase (NOS) family, which synthesises nitric oxide (NO) in endothelial and neuronal cells. NO synthesis can be inhibited by endogenous asymmetric dimethylarginine (ADMA). NO has explicit roles in cellular signalling and vasodilation. Impaired NO bioavailability represents the central feature of endothelial dysfunction associated with vascular diseases. Interestingly, dietary supplementation with *L*-arginine has been shown to alleviate endothelial dysfunctions caused by impaired NO synthesis. In this study the transport kinetics of [3H]-arginine and [3H]-ADMA into the central nervous system (CNS) were investigated using physicochemical assessment and the *in situ* brain/choroid plexus perfusion technique in anesthetized mice. Results indicated that *L*-arginine and ADMA are tripolar cationic amino acids and have a gross charge at pH 7.4 of 0.981. *L*-Arginine (0.00149±0.00016) has a lower lipophilicity than ADMA (0.00226±0.00006) as measured using octanol-saline partition coefficients. The *in situ* perfusion studies revealed that [3H]-arginine and [3H]-ADMA can cross the blood-brain barrier (BBB) and the blood-CSF barrier. [3H]-Arginine (11.6nM) and [3H]-ADMA (62.5nM) having unidirectional transfer constants ($K_{in}$) into the frontal cortex of 5.84±0.86 and 2.49 ±0.35 µl.min⁻¹.g⁻¹, respectively, and into the CSF of 1.08±0.24 and 2.70±0.90 µl.min⁻¹.g⁻¹, respectively. In addition, multiple-time uptake studies revealed the presence of CNS-to-blood efflux of ADMA. Self- and cross-inhibition studies indicated the presence of transporters at the BBB and the blood-CSF barriers for both amino acids, which were shared to some degree. Importantly, these results are the first to demonstrate: (i) saturable transport of [3H]-ADMA at the blood-CSF barrier (choroid plexus) and (ii) a significant CNS to blood efflux of [3H]-ADMA. Our results suggest that the arginine paradox, in other words the clinical observation that NO-deficient patients respond well to oral supplementation with *L*-arginine even though the plasma concentration is sufficient to saturate endothelial NOS, could be related to altered ADMA transport (efflux).

**Data Availability Statement:** All relevant data are within the manuscript and its Supporting Information files.

**Funding:** This work was supported by a Biotechnology and Biological Sciences Research Council (BBSRC) centre for integrative biomedicine PhD studentship for Mr Fidanboylu to Dr Sarah Ann Thomas [BB/E527098/1]. https://www.ukri.org/councils/bbsrc/. This research was funded in whole, or in part, by the Wellcome Trust [080268]. https://wellcome.org/. The recipients of this Wellcome grant were: Dr Sarah Ann Thomas, Professors Paul Thomas Francis, Marzia Malcangio, Stephen Brendan McMahon and Marcus Rattray. For the purpose of Open Access, the author has applied a CC BY public copyright licence to any Author Accepted Manuscript version arising from this submission. The funders had no role in study design, data collection and analysis, decision to publish, or preparation of the manuscript.

**Competing interests:** The authors have declared that no competing interests exist.

**Abbreviations:** ADMA, Asymmetric dimethylarginine or $N^G$,$N^G$-dimethyl-*L*-arginine; BBB, blood-brain barrier; CAT1, cationic amino acid transporter-1; CNS, central nervous system; CSF, cerebrospinal fluid; CVOs, circumventricular organs; DDAH, dimethylarginine dimethylaminohydrolase; dpm, disintegrations per minute; eNOS, endothelial nitric oxide synthase; $K_m$, half-saturation constant; hCMEC/D3, human cerebral microvessel endothelial cells; $V_i$, initial volume of distribution; NO, nitric oxide; NOS, nitric oxide synthase; $K_{in}$, unidirectional transfer constant.

## Introduction

Nitric oxide (NO) is implicated in cerebral microvessel basal tone, blood flow autoregulation and endothelium junctional integrity [1, 2]. In the brain, NO participates in cell signalling and in pathogen defence, and disordered NO generation is associated with pathological conditions such as Alzheimer's disease [3]. NO is synthesised from *L*-arginine by nitric oxide synthase (NOS) enzymes. Three human isoforms of NOS have been characterised [4]; which are classified according to the tissue or cell type of origin when determining cDNA expression. Two of the isoforms—endothelial (eNOS) and neuronal (nNOS), are constitutively expressed, while the third isoform is inducible (iNOS). Asymmetric dimethylarginine (ADMA) is produced by all cells and is a product of protein degradation and competes with *L*-arginine for each of the three NOS isoforms [5]. However, ADMA cannot be used as a substrate by NOS and as a result ADMA is a biologically significant inhibitor of NO production [6, 7].

As *L*-arginine is the precursor for NO synthesis [8] and ADMA a potent endogenous inhibitor of NO synthesis [6, 7], the arginine/ADMA plasma concentration ratio is widely considered to be an important indicator of NO bioavailability [9, 10]. In addition, *L*-arginine and ADMA have been shown to influence their human brain endothelial and umbilical vein endothelial cellular availability via transport interactions [11, 12]. Together these facts imply that the interplay between the transport of these two cationic amino acids at the blood-brain barrier (BBB) and blood-cerebrospinal fluid (CSF) barriers is likely to directly relate to NO production at these interfaces and within the central nervous system (CNS). Interestingly, dietary supplementation with *L*-arginine has been shown to alleviate endothelial dysfunctions caused by impaired NO synthesis [13, 14], despite eNOS already being saturated with *L*-arginine [15, 16]. This is called the arginine paradox. Taking into consideration that ADMA can enter and exit cells [17, 18], understanding ADMA and *L*-arginine transport in more detail at the blood-brain and blood-CSF barriers will contribute to further understanding the pathway for NO generation and how it is potentially altered by disease and supplements. We have already investigated [3H]-arginine and [3H]-ADMA transport using human cerebral microvessel endothelial cells (hCMEC/D3) cells, which is an established *in vitro* model of the BBB [12]. In this present study we (i) utilized a chemical property database to compare the physicochemical characteristics of the cationic amino acids and (ii) examined the transport of [3H]-arginine and [3H]-ADMA using an *in situ* brain /choroid plexus perfusion method in anaesthetized mice. This sensitive method allows the transport processes at the BBB (so the brain capillary wall) to be distinguished from those at the blood-CSF interface (so the choroid plexuses and the arachnoid membrane) [19, 20]. We will also explore the hypothesis that ADMA efflux could be responsible for the increase in NO production after arginine supplementation by using the *in situ* method to perform self- and cross-inhibitor transport studies.

## Methods

### Materials

[3H]-arginine (mol. wt., 174.2 g/mol; specific activity, 43 Ci/mmol; >97% radiochemical purity) was purchased from Amersham Radiochemicals, Buckinghamshire, UK. [3H]-ADMA (mol. wt., 276.7 g/mol; specific activity, 8 Ci/mmol; 96.4% radiochemical purity) was synthesized and tritiated by Amersham Radiochemicals, Cardiff, UK. [14C]-sucrose (mol. wt., 342.3 g/mol; specific activity, 0.412 Ci/mmol; 99% radiochemical purity; Moravek Biochemicals, Brea, CA). Unlabelled *L*-arginine and $N^G$,$N^G$-dimethylarginine dihydrochloride (ADMA) were purchased from Sigma Aldrich (Dorset, UK).

## Physicochemical characteristics

The physicochemical properties of *L*-arginine, ADMA and sucrose were obtained from the chemical properties database, MarvinSketch [21]. These properties included their chemical structures, the percentage distribution of the different microspecies at physiological pH, molecular weight, predicted log D at pH 7.4 and the gross charge distribution at pH 7.4.

The lipophilicity of the radiolabelled test molecules was also measured by means of an octanol:saline partition coefficient. 0.75 mL phosphate buffered saline (pH 7.4) containing 1 μCi (0.037 MBq) of either [$^{14}$C]-sucrose (2.5 μM), [$^3$H]-arginine (101.0 nM) or [$^3$H]-ADMA (542.8 nM) was mixed with 0.75 mL octanol (Sigma Aldrich; Dorset, UK). Centrifugation separated the hydrophobic (octanol) phase and hydrophilic (saline) phase and 100 μL samples in triplicate were taken for radioactive liquid scintillation counting on a Tricarb 2900TR liquid scintillation counter (Perkin-Elmer; Boston, MA, USA). The partition coefficient was calculated as the ratio of radioactivity in the octanol phase to the radioactivity in the aqueous phase. The coefficients for *L*-arginine and ADMA were compared.

## Animals & anaesthesia

All experiments requiring the use of animals were performed in accordance with the Animal (Scientific Procedures) Act, 1986 and Amendment Regulations 2012 and with consideration to the Animal Research: Reporting of *In Vivo* Experiments (ARRIVE) guidelines. The study was approved by the King's College London Animal Welfare and Ethical Review Body and performed under license: 70/6634.

All animals used in procedures were adult male BALB/c mice (between 23 g and 25 g) sourced from Harlan Laboratories, Oxon, UK, unless otherwise stated. Animals were handled with due care and consideration to minimise stress. All animals were maintained under standard temperature/lighting conditions and given water and food *ad libitum*. Welfare was monitored daily by animal technicians. Animals which were showing signs of distress (such as not feeding or drinking) were brought to the attention of the named veterinary surgeon and the animal euthanized. Mice were used for all the procedures under a non-recovery anaesthetic and represent the animals of lowest neurological sensitivity to which

h the protocols can be successfully applied. Animals were weighed immediately before the administration of the anaesthetic to ensure correct dosage. Domitor® (medetomidine hydrochloride) and Vetalar® (ketamine) were both purchased from Harlan Laboratories (Cambridge, UK). All animals were terminally anaesthetised (2 mg/kg Domitor® and 150 mg/kg Vetalar® injected intraperitoneally), and a lack of self-righting and paw-withdrawal reflexes (as surrogate indicators of consciousness) thoroughly checked prior to and during all procedures. If needed, additional anaesthetic was administered via the same route. During the general anaesthesia no adverse effects were expected or observed. At the end of the *in situ* brain perfusion procedure the animal was killed by decapitated under non-recovery anaesthesia. Animals were heparinised with 100 units heparin (in 0.9% m/v NaCl$_{(aq)}$, Harlan Laboratories; Oxon, UK), administered *via* the intraperitoneal route prior to surgery. Experiments were performed between 9 am and 5 pm.

## *In situ* brain/choroid plexus perfusion

The *in situ* brain/choroid plexus perfusion method is designed to directly perfuse the brain with artificial plasma *via* a cannula placed in the heart. Artificial plasma is prepared to mimic the ionic composition of blood, and transporter inhibitors can be added in known concentrations and delivered to the brain for defined periods of time. There are several advantages associated with using the *in situ* brain/choroid plexus perfusion technique over other methods (e.g.

brain uptake index method) to study blood-brain and blood-CSF barrier transport. These advantages include:

i.  Perfusions can be carried out over a long period of time (up to 30 minutes), which allows measurement of the CNS uptake of slowly permeating molecules [20, 22].

ii.  Heart perfusions allow transport across the BBB and blood-CSF barrier to be measured simultaneously [20].

iii.  The perfusion flow rate is controlled by means of a peristaltic pump. Thus the solute of interest is delivered to the CNS at a constant flow and a known concentration that does not vary with time and the kinetics of transport can be calculated accordingly [20, 22].

iv.  The concentration of transporter inhibitors in the artificial plasma can easily be manipulated to measure saturable transport across the BBB and blood-CSF barrier and identify the transport systems involved [22].

v.  The radiolabelled and unlabelled solutes are in an artificial plasma that does not contain enzymes and are exposed to the brain microcirculation before any peripheral organs involved in metabolism (*e.g.* liver, kidneys or lungs). Thus one can be confident that the compound is intact and the radiolabel is attached when interacting with transporters at the blood-brain and blood-CSF barriers [22].

**Surgical preparation.**   The experimental procedure is well established [20]. The left ventricle of the heart was cannulated with a 25G x 10 mm butterfly-winged needle connected to the perfusion circuit. The artificial plasma was warmed to 37˚C and oxygenated (95% $O_2$ and 5% $CO_2$) before being perfused into the heart *via* this circuit at a flow rate of 5 mL/min. The right atrium of the heart was sectioned to allow outflow of the artificial plasma. Thus, an open circuit is created and the artificial plasma only passes through the circulation once. The artificial plasma consisted of a modified Krebs-Henseleit mammalian Ringer solution with the following constituents: 117 mM NaCl, 4.7 mM KCl, 2.5 mM $CaCl_2$, 1.2 mM $MgSO_4$, 24.8 mM $NaHCO_3$, 1.2 mM $KH_2PO_4$, 10 mM glucose, and 1 g/liter bovine serum albumin. Radiolabelled and unlabelled test molecules were infused into the inflowing artificial plasma for periods up to 30 minutes.

**CNS sampling.**   At the set perfusion time a CSF sample was taken by inserting a pulled glass micropipette connected to silicon tubing and a syringe into the cisterna magna. Gentle suction was applied, and CSF was withdrawn. Only clear CSF samples were processed and taken for analysis. Samples contaminated with blood were discarded. The animal was then decapitated and the brain removed. Specific brain samples including the circumventricular organs (CVOs) were taken using a Leica S4E L2 stereomicroscope (Leica; Buckinghamshire, UK). The brain samples included the frontal cortex, caudate nucleus, occipital cortex, hippocampus, hypothalamus, thalamus, pons and cerebellum. The CVOs included the choroid plexus, pituitary gland and pineal gland.

**Capillary depletion analysis.**   The microvasculature within a sample of brain tissue can be separated from the brain parenchyma using capillary depletion analysis [23]. Following perfusion, the remaining brain tissue after microdissection of the regions described above (typically 200–300 mg) underwent capillary depletion following a method modified for mouse [20]. A whole-brain homogenate was prepared using physiological capillary depletion buffer (141 mM NaCl, 4.0 mM KCl, 2.8 mM $CaCl_2$, 1.0 mM $MgSO_4$, 10.9 mM HEPES, 1.0 mM $NaH_2PO_4$, 10 mM glucose at pH 7.4) and a manual homogeniser, before adding 26% m/v dextran (final concentration 13% m/v, 60-90kDa, MP Biomedicals Europe, France) and homogenising again. Both the physiological capillary depletion buffer and dextran solution were maintained at 4˚C

both to halt any cell metabolism and transport processes, and to reduce any cellular or protein damage due to heat generated from the homogenisation process. The homogenate was centrifuged at 5,400 r.c.f and 4˚C to produce an endothelial cell-enriched pellet and supernatant containing brain parenchyma.

**Liquid scintillation analysis.** All samples (brain regions, capillary depletion brain homogenate, pellet and supernatant, CVOs, CSF and plasma samples) were solubilised in 0.5 mL tissue solubiliser (Solvable; Perkin-Elmer; Boston, MA, USA). After incubating the samples at room temperature for 48 h, 4 mL scintillation fluid (Lumasafe®; Perkin-Elmer; Boston, MA, USA) was added to each before vigorous vortexing. The amounts of [3H] and [14C] radioactivity in each sample were then quantified using a Packard Tri-Carb 2900TR counter (Perkin-Elmer; Boston, MA, USA). Counts per minute were then converted to disintegrations per minute (dpm) by the counter using internally stored quench curves from standards and corrected for background dpm.

## Experiment design

The *in situ* brain perfusion experiments contained [3H]-arginine (11.6 nM) or [3H]-ADMA (62.5 nM) and [14C]sucrose in the artificial plasma. Multiple-time (2.5, 10, 20 and 30 minute) uptake studies were performed to measure the uptake of [3H]-arginine or [3H]-ADMA into the mouse CNS over time. Self-inhibition experiments were also performed. In these experiments the perfusion period was 10 minutes and [3H]-arginine was measured in the presence of 100 μM unlabelled *L*-arginine and the uptake of [3H]-ADMA measured in the presence of 100 μM unlabelled ADMA. Cross-competition experiments were also performed. In these experiments the perfusion period was 10 minutes and [3H]-arginine was measured in the presence of 0.5–500 μM unlabelled ADMA and the uptake of [3H]-ADMA measured in the presence of 100 μM arginine.

[14C]-Sucrose is used as an internal control and determines blood-brain and blood-CSF barrier integrity in each experiment. It can also provide a measure of the cerebrovascular space in different regions, and the extracellular space formed between choroid plexus capillary endothelium and epithelium [19]. In the pituitary gland and pineal gland samples, [14C]-sucrose represents the vascular space and the ability of [14C]-sucrose to cross between capillary endothelial cells in the absence of tight junctions.

Values for uptake of [3H]-arginine or [3H]-ADMA in brain regions can be corrected for [14C]-sucrose vascular space to standardize uptake and reduce regional discrepancies due to differences in vascularity. It also corrects for individual variation. Thus, any differences in the brain uptake of radiolabelled solutes are likely to be due to differences in transporter expression, rather than due to differences in blood supply.

## Expression of results

The concentration of radioactivity in the brain regions ($C_{BRAIN}$; dpm/g), CVOs ($C_{CVO}$; dpm/g) and CSF ($C_{CSF}$; dpm/mL) was expressed as a percentage of that in the artificial plasma ($C_{pl}$; dpm/mL) and termed $R_{BRAIN}$ (dpm/100g), $R_{CVO}$ (dpm/100g) or $R_{CSF}$ (dpm/100mL), respectively, or $R_{Tissue}$ (dpm/100g) as shown in Eq 1:

$$R_{Tissue} = \frac{C_{Tissue}}{C_{pl}} \times 100 \qquad\qquad \text{Eq1}$$

Correcting for vascular space involved subtracting the $R_{Tissue}$ value obtained for [14C]-sucrose in each sample from the $R_{Tissue}$ value concurrently obtained for the [3H]-labelled cationic amino acid.

Using these data, the permeability of the BBB and blood-CSF barrier to the solute of interest can be calculated as a unidirectional transfer constant ($K_{in}$). There are two methods to determine the $K_{in}$ value:

i. Single time point analysis method, described by Eq 2

ii. Multiple time point analysis method, described by Eq 3

The first method: (i) was developed by [24, 25] and applied to the brain perfusion data [26].

$$K_{in} = \frac{C_{Tissue}(T)}{C_{pl}T} \qquad \text{Eq2}$$

Where $C_{Tissue}$ (T) is radioactivity (dpm) per g of tissue at time-point T (perfusion time in minutes), and $C_{pl}$ is radioactivity (dpm) per mL of artificial plasma.

Eq 2 assumes that the entry of the radiolabelled solute of interest into the

CNS is proportional to, but less than, its concentration in the artificial plasma and efflux (CNS to blood) is much smaller than influx (blood to CNS) of the test solute and therefore can be ignored [27].

It should however be noted that calculating the $K_{in}$ value from blood to CNS using this method requires that $R_{Tissue}$ at time T is first corrected for vascular space by subtracting the $R_{Tissue}$ value for [$^{14}$C]-sucrose determined at that time-point.

The second method (ii) for calculating the $K_{in}$ value from blood to CNS requires multiple-time uptake data and Eq 3 [26, 28–30]:

$$\frac{C_{Tissue}(T)}{C_{pl}(T)} = K_{in}T + V_i \qquad \text{Eq3}$$

Where $C_{Tissue}$ (T) and $C_{pl}$ (T) are radioactivities per unit weight of tissue and plasma at time T; T being the length of perfusion. $V_i$ is the initial volume of distribution of the test solute in the rapidly equilibrating space (which may include the vascular space, the capillary endothelial volume and /or compartments in parallel with the BBB, such as the space between the brain capillary endothelium and the neuroglial lining of the capillary) [26, 27, 31]. Thus, plotting the data points resulting from **Eq 1** at each time point defines a straight line ($y = mx+c$), where $K_{in}$ is the slope, and $V_i$ is the ordinate intercept. Any transport of the radiolabelled solute of interest back from the brain to the blood can be observed by a loss of linearity of the experimental points [26]. During the experimental period when Eq 3 is applicable, the amount of test substance in $V_i$ is roughly proportional to $C_{pl}$ and the test substance moves unidirectionally from plasma into brain tissue [31].

The percentage change in the $R_{Tissue}$ uptake values achieved in the absence or presence of an unlabelled inhibitor can be calculated by means of Eq 4.

$$\%change = \frac{R_{Tissue} - R_{Inhibition}}{R_{Tissue}} \times 100 \qquad \text{Eq4}$$

We define $R_{Inhibition}$ as the $R_{Tissue}$ uptake in the presence of an inhibitor in the artificial plasma.

A decrease in the uptake of radiolabelled solute in the presence of unlabelled inhibitor is indicative of a saturable influx transport system. Conversely, an increase in the uptake of radiolabelled solute in the presence of unlabelled inhibitor is indicative of a saturable efflux transport system. No change in the distribution of the radiolabelled solute in the absence or presence of the unlabelled inhibitor may indicate the absence of saturable transport by the radiolabelled solute or the use of influx and efflux transporters by the radiolabelled solute.

## Statistics

Data from all experiments are presented as mean ± standard error of the mean (SEM).

The following statistical tests were applied as specified in the following sections: unpaired Student's t-test, paired Student's t-test, One-way ANOVA with Dunnett's multiple comparison of means or two-way ANOVA with Tukey's multiple comparison test as appropriate. Statistical significance was taken as follows: not significant (ns), $p > 0.05$, $^*p < 0.05$, $^{**}p < 0.01$, $^{***}p < 0.001$. All statistical analyses were performed using GraphPad Prism v5.0c or v6 graphing and statistics package for Mac.

## Results

### Physicochemical characteristics

The physicochemical characteristics of *L*-arginine and ADMA were obtained from the chemical properties database, MarvinSketch [21]. *L*-Arginine and ADMA are cationic (tripolar) amino acids, which exist as two microspecies at physiological pH. The percentage distribution of the two microspecies for each amino acid is shown in Fig 1. The major microspecies of both *L*-arginine (98.15%) and ADMA (98.11%) has a positive charge (+1). The minor microspecies of both *L*-arginine (1.85%) and ADMA (1.88%) are zwitterions. Both *L*-arginine and ADMA have a gross charge at physiological pH of +0.981. The molecular weights of arginine and ADMA are 174.2 g/mol and 202.26 g/mol respectively. The log D at pH 7.4 of arginine is -4.77 and ADMA is -3.99 [21]. Octanol-saline partition coefficients at pH 7.4 of [³H]-arginine and [³H]-ADMA were also measured as part of this study and were found to be 0.00149±0.00016 and 0.00226±0.00006, respectively.

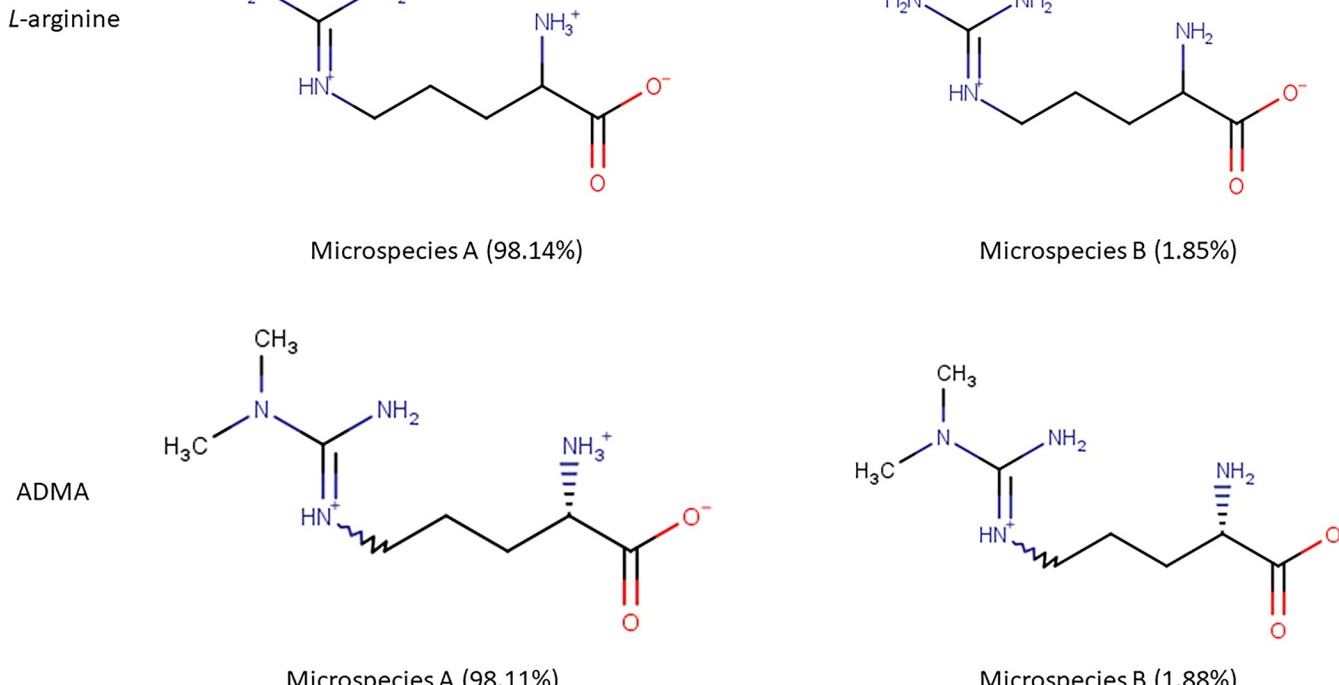

**Fig 1. The percentage distribution and chemical structures of the two *L*-arginine microspecies and the two ADMA microspecies found at physiological pH.** Microspecies A is the major microspecies. Microspecies B is the minor microspecies.

[$^{14}$C]-Sucrose is used as a baseline marker molecule. It has a molecular weight of 342.3 g/mol and it has a log D at pH 7.4 of -4.87 [21]. The octanol-saline partition coefficient at pH 7.4 of [$^{14}$C]-sucrose was 0.00105±0.00022.

## Arginine

**Brain distribution.** Fig 2 shows the distribution ($R_{BRAIN}$) of the vascular space marker molecule, [$^{14}$C]-sucrose, and the cationic amino acid, [$^{3}$H]-arginine, into different regions of the brain over time as measured by the *in situ* brain/choroid plexus perfusion method. Multiple time uptake analysis of [$^{14}$C]-sucrose into the different brain regions was used to calculate a unidirectional transfer constant ($K_{in}$) and an initial volume of distribution ($V_i$) by means of Eq 3 and the values are reported in Table 1. For example, the $K_{in}$ was 0.070±0.016 µl.min$^{-1}$.g$^{-1}$ and the $V_i$ was 1.40±0.31 mL.100g$^{-1}$ for [$^{14}$C]-sucrose distribution into the frontal cortex. The low distribution of [$^{14}$C]-sucrose into all regions of the brain and at all time points after *in situ* perfusion confirms the integrity of the BBB in each experiment.

The uptake of [$^{3}$H]-arginine was significantly greater than [$^{14}$C]-sucrose in all brain regions and at all-time points (multiple Student's paired t-tests, $p < 0.05$ in all cases) (Fig 2). A time-dependent increase in the distribution of [$^{3}$H]-arginine (corrected for [$^{14}$C]-sucrose) was observed in all regions (*e.g.* 28.96±4.03% after 2.5 minutes to 176.78±31.60% after 30 minutes in the frontal cortex) and no regional differences were observed ($p > 0.05$, two-way ANOVA with Tukey's multiple comparison test for each brain region compared to all others within each time point).

Multiple-time uptake analysis over 30 minutes could not be used to calculate a $K_{in}$ and $V_i$ for [$^{3}$H]-arginine samples, as Eq 3 is only applicable when the amount of substance in $V_i$ is roughly proportional to $C_{pl}$. The $V_i$ values determined for [$^{3}$H]-arginine using this method were up to 13-fold higher than that measured for [$^{14}$C]-sucrose space in the same experiment

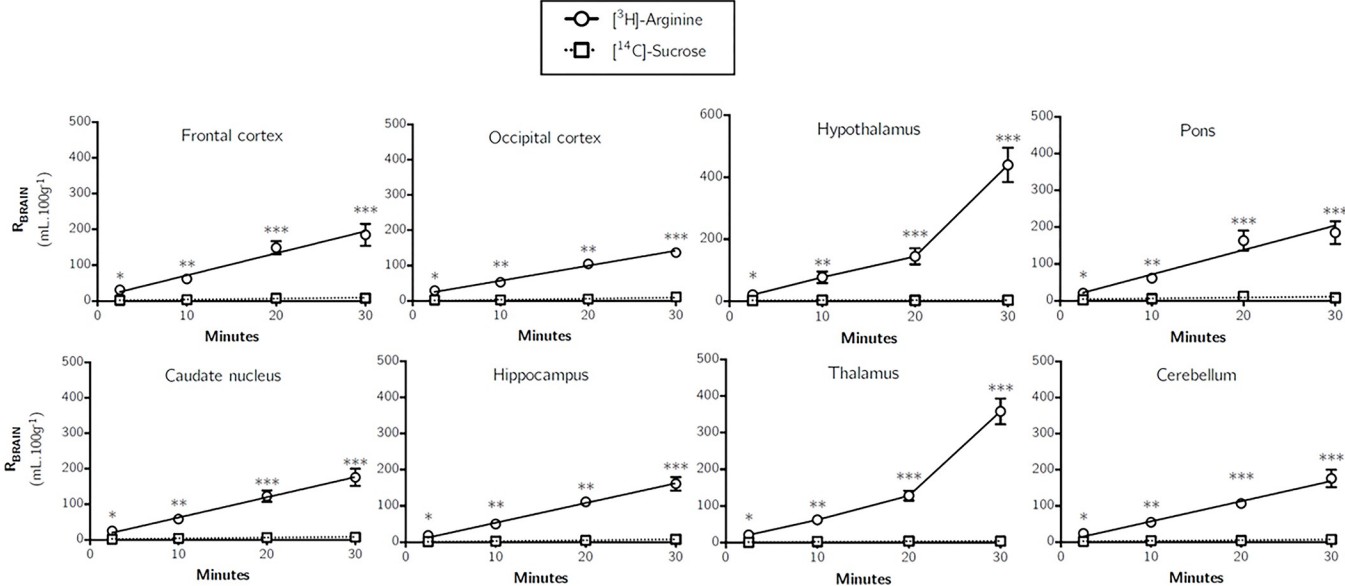

**Fig 2. The brain distribution of [$^{3}$H]-arginine and [$^{14}$C]-sucrose as a function of time measured using the *in situ* brain perfusion technique in anaesthetised mice.** Uptake is expressed as the percentage ratio of tissue to plasma (mL.100 g$^{-1}$). Each point represents the mean ± SEM of 4–7 animals. $K_{in}$ and $V_i$ values were determined as the slope and ordinate intercept of the computed regression lines where appropriate and reported in Table 1. Asterisks represent one-tailed, paired Student's t-tests comparing mean±SEM at each time point, *$p < 0.05$, **$p < 0.01$, ***$p < 0.001$ (GraphPad Prism 6.0 for Mac).

**Table 1. The unidirectional transfer constants ($K_{in}$) and initial volume of distribution ($V_i$) determined by multiple-time uptake analysis of [$^{14}$C]-sucrose into specific CNS regions.**

| Brain Regions | *p* | $R^2$ | $K_{in}$ µl.min$^{-1}$.g$^{-1}$ | $V_i$ mL.100g$^{-1}$ |
|---|---|---|---|---|
| Frontal Cortex | 0.0004 | 0.51 | 0.070±0.016 | 1.40 ± 0.31 |
| Caudate Nucleus | 0.0005 | 0.50 | 0.067±0.016 | 1.19±1.19 |
| Occipital Cortex | <0.0001 | 0.61 | 0.068±0.013 | 1.16±0.24 |
| Hippocampus | <0.0001 | 0.66 | 0.060±0.010 | 0.81±0.19 |
| Hypothalamus | 0.0491 | 0.82 | 0.044±0.023 | 2.67±0.43 |
| Thalamus | 0.0001 | 0.57 | 0.107±0.022 | 0.87±0.41 |
| Pons | 0.0074 | 0.35 | 0.086±0.028 | 2.52±0.51 |
| Cerebellum | 0.0026 | 0.40 | 0.058±0.017 | 1.66±0.31 |
| **Capillary depletion analysis** | | | | |
| Homogenate | 0.0061 | 0.38 | 0.061±0.019 | 1.51±0.38 |
| Supernatant | 0.0006 | 0.55 | 0.058±0.013 | 0.81±0.27 |
| Pellet | 0.0151 | 0.36 | 0.0069±0.0025 | 0.21±0.051 |

Linear regression analysis produced $R^2$ for goodness of fit and *p*-values relating to the deviation of the line slope from zero. There was no significant difference between the [$^{14}$C]-sucrose values reported for the perfusion studies with [$^3$H]-arginine (data shown) and for those reported with [$^3$H]-ADMA (data not shown).

(data not shown). However, single-time uptake analysis (Eq 2) to calculate a transfer constant ($K_{in}$) for [$^3$H]-arginine distribution into all brain regions could be used and the values are reported in Table 2. For example, single-time uptake analysis of [$^3$H]-arginine at 10 minutes into the frontal cortex, hypothalamus and thalamus revealed a $K_{in}$ of 5.84±0.86, 7.26±1.76 and 5.87±1.07 µl.min$^{-1}$.g$^{-1}$, respectively (Table 2).

**Table 2. The *Kin* values for [$^3$H]-arginine and [$^3$H]-ADMA values determined by single-time point analysis at a 10-minute perfusion.**

| Brain Region | [$^3$H]-arginine $K_{in}$ (µl.min$^{-1}$.g$^{-1}$) | n | [$^3$H]-ADMA $K_{in}$ (µl.min$^{-1}$.g$^{-1}$) | n | [$^3$H]-ADMA/ [$^3$H]-arginine % |
|---|---|---|---|---|---|
| Frontal Cortex | 5.84±0.86 | 7 | 2.49±0.35 | 5 | 42.7 |
| Caudate nucleus | 5.40±0.67 | 7 | 2.41±0.29 | 5 | 44.6 |
| Occipital Cortex | 4.98±0.72 | 7 | 2.25±0.16 | 5 | 45.2 |
| Hippocampus | 4.72±0.56 | 7 | 2.17±0.17 | 5 | 46.1 |
| Hypothalamus | 7.26±1.76 | 7 | 3.04±0.30 | 5 | 41.9 |
| Thalamus | 5.87±1.07 | 7 | 2.10±0.17 | 5 | 35.8 |
| Pons | 5.58±0.97 | 7 | 2.46±0.34 | 5 | 44.2 |
| Cerebellum | 5.10±0.50 | 7 | 1.62±0.19 | 5 | 31.7 |
| **Capillary depletion analysis** | | | | | |
| Homogenate | 3.97±0.67 | 7 | 1.27±0.56 | 5 | 31.9 |
| Supernatant | 2.35±0.42 | 7 | 0.86±0.16 | 5 | 36.6 |
| Pellet | 0.70±0.10 | 7 | 0.08±0.20 | 5 | 11.4 |
| **Circumventricular organs** | | | | | |
| CSF | 1.08±0.24 | 5 | 2.70±0.90 | 5 | 249.5 |
| Choroid plexus | Not applicable | - | Not applicable | | - |
| Pituitary gland | Not applicable | - | Not applicable | | - |
| Pineal gland | Not applicable | - | Not applicable | | - |

The $R_{Tissue}$ values had all been corrected for [$^{14}$C]-sucrose. The rate of uptake of [$^3$H]-ADMA is expressed as a percentage of the rate of uptake of [$^3$H]-arginine.

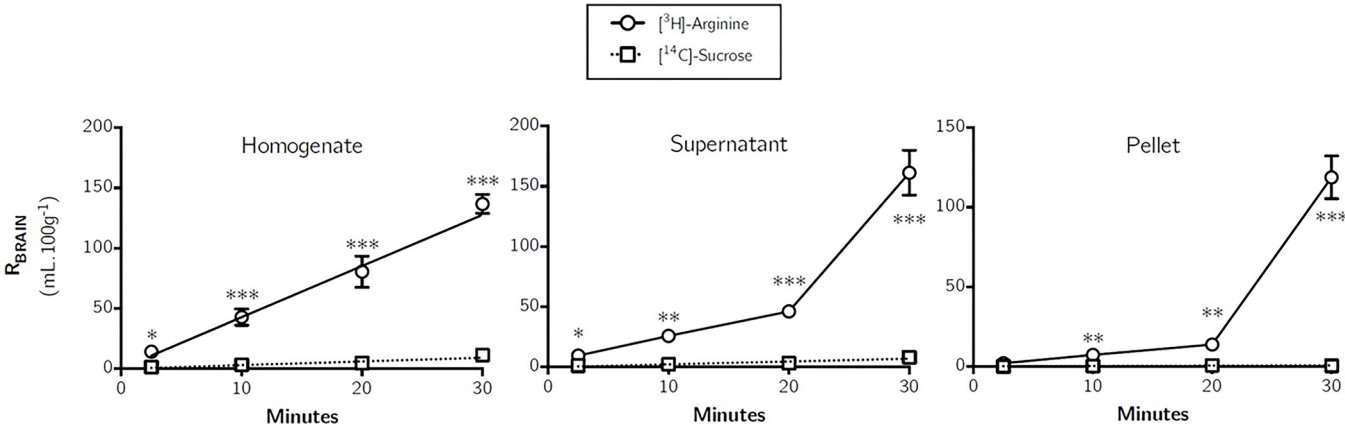

**Fig 3. Distribution of [³H]-arginine and [¹⁴C]-sucrose in capillary depletion samples as a function of time.** Uptake is expressed as the percentage ratio of tissue to plasma (mL.100 g⁻¹). Each point represents the mean ±SEM of 4–7 animals. (GraphPad Prism 6.0 for Mac). Asterisks represent one-tailed, paired Student's t-tests comparing mean±SEM at each time point, $^*p < 0.05$, $^{**}p < 0.01$, $^{***}p < 0.001$.

Fig 3 confirms the presence of [³H]-arginine in the brain homogenate, brain parenchyma-containing supernatant and endothelial cell enriched pellet. There was an increase in the distribution of [³H]-arginine in all three capillary depletion samples over time (Fig 3). There was a significantly higher uptake of [³H]arginine than [¹⁴C]sucrose in the brain homogenate, brain supernatant and capillary endothelial cell enriched pellet at all time points, except at 2.5 minutes for the pellet sample (one-tailed, paired Student's t-tests) (Fig 3). No differences in the distribution of [³H]-arginine between the homogenate and supernatant (containing brain parenchyma) were observed, however both the homogenate and supernatant samples contained a higher distribution than that observed in the endothelial cell-enriched pellet ($p > 0.01$, two-way ANOVA with Tukey's multiple comparison test for each sample compared to all others).

**CSF.** Fig 4 shows that the [³H]-arginine uptake into the CSF was 4.41±1.41% at 2.5 minutes and rose to 120.44±25.08% at 30 minutes (data uncorrected for [¹⁴C]-sucrose). The K$_{in}$ determined by single-time uptake analysis for [³H]-arginine into the CSF at 10 minutes was 1.08±0.24 µl.min⁻¹.g⁻¹ (Table 2).

**Circumventricular organs.** Fig 4 illustrates the distribution of [¹⁴C]-sucrose and [³H]-arginine into the choroid plexus. The [¹⁴C]-sucrose distribution represents the extracellular

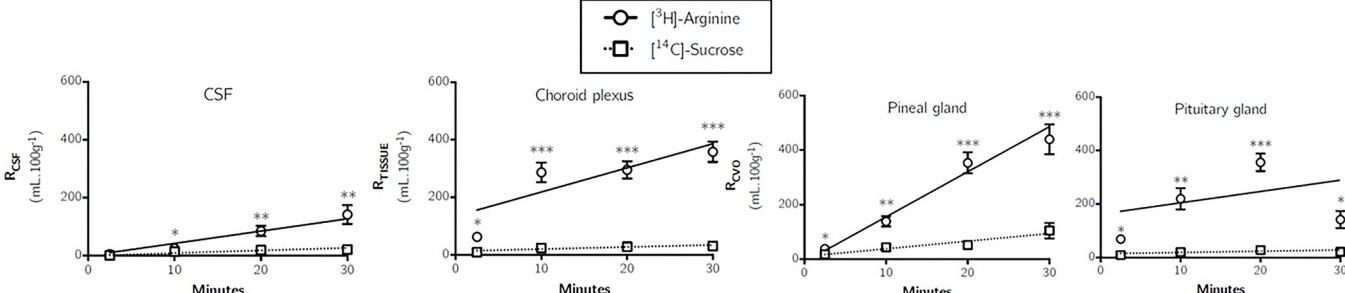

**Fig 4. Distribution of [³H]-arginine and [¹⁴C]-sucrose in the CSF, pineal gland, choroid plexus and pituitary gland following *in situ* brain perfusion as a function of time.** Uptake is expressed as the percentage ratio of tissue or CSF to plasma (mL.100 g⁻¹). Each point represents the mean ± SEM of 4–7 animals. Asterisks represent one-tailed, paired Student's t-tests comparing mean±SEM at each time point, $^*p < 0.05$, $^{**}p < 0.01$, $^{***}p < 0.001$ (GraphPad Prism 6.0 for Mac).

space formed between choroid plexus capillary endothelium and epithelium in this highly vascularised tissue. [3H]-Arginine accumulation was significantly greater than [14C]-sucrose accumulation at all time-points (one-tailed, paired Student's t-tests comparing mean±SEM at each time point).

Fig 4 illustrates the distribution of [14C]-sucrose and [3H]-arginine into the pituitary gland and pineal gland. [3H]-Arginine accumulation was significantly greater than [14C]-sucrose accumulation at all time-points in both tissues (one-tailed, paired Student's t-tests comparing mean±SEM at each time point). A comparison of the mean [3H]-arginine distributions between the pituitary gland and pineal gland revealed that a significant difference is observed between the pituitary gland and pineal gland, but only after a 30-minute perfusion (p<0.001, two way ANOVA with Tukey's multiple comparison test for each sample compared to all others within each time point). Single-time uptake analysis could not be used to determine a $K_{in}$ for [3H]-arginine transfer into the choroid plexus, pituitary gland and pineal gland as the [3H]-arginine concentration in these tissues was higher than the concentration in the plasma (Table 2). Single-time uptake analysis to calculate a transfer constant can only be applied if entry of the test solute into the CNS is proportional to its plasma concentration and the concentration in the CNS is less than the concentration in the plasma and efflux (CNS to blood) is much smaller than influx (blood to CNS) of the test solute and therefore can be ignored [27].

## ADMA

**Brain distribution.** Fig 5 shows the distribution ($R_{Brain}$) of [14C]-sucrose and [3H]-ADMA into the frontal cortex, caudate nucleus, occipital cortex, hippocampus, hypothalamus, thalamus, pons and cerebellum over time. The uptake of [14C]-sucrose was low in all brain regions confirming the integrity of the BBB. The uptake of [14C]-sucrose was significantly lower than [3H]-ADMA in all brain regions and at all-time points (multiple Student's paired t-tests, $p < 0.05$ in all cases).

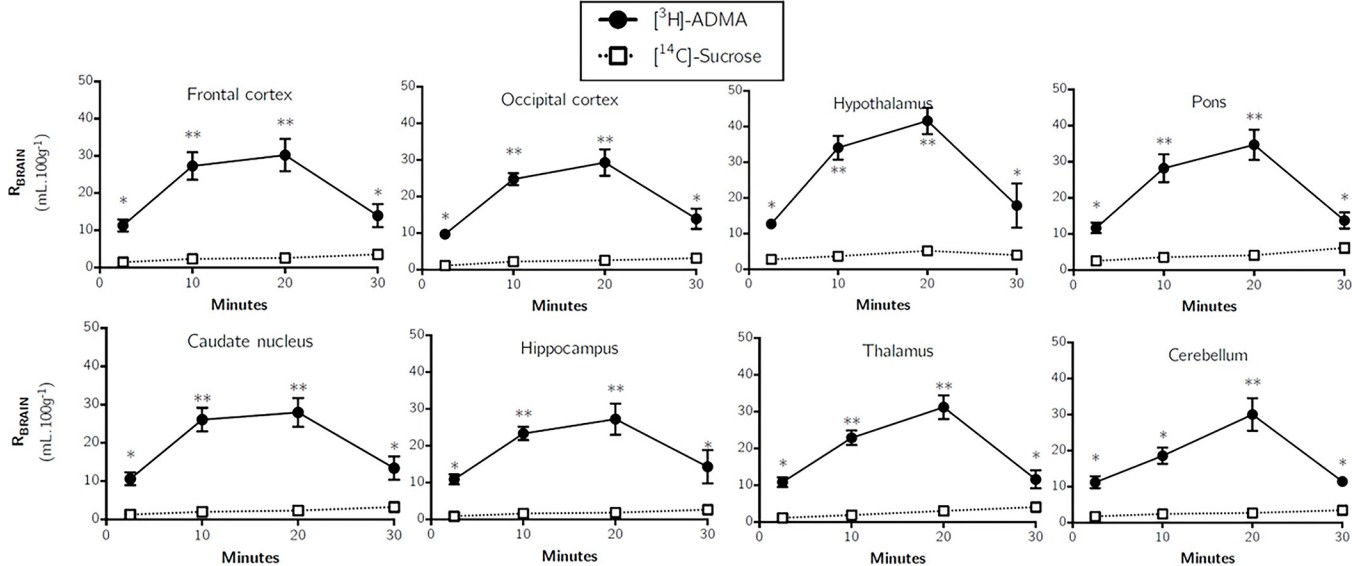

**Fig 5. Brain distribution of [3H]-ADMA and [14C]-sucrose as a function of time.** Uptake is expressed as the percentage ratio of tissue to plasma (mL.100 g$^{-1}$). Each point represents the mean ± SEM of 5 animals. One-tailed, paired Student's t-tests comparing mean±SEM at each time point, $^{*}p < 0.05$, $^{**}p < 0.01$ (GraphPad Prism 6.0 for Mac).

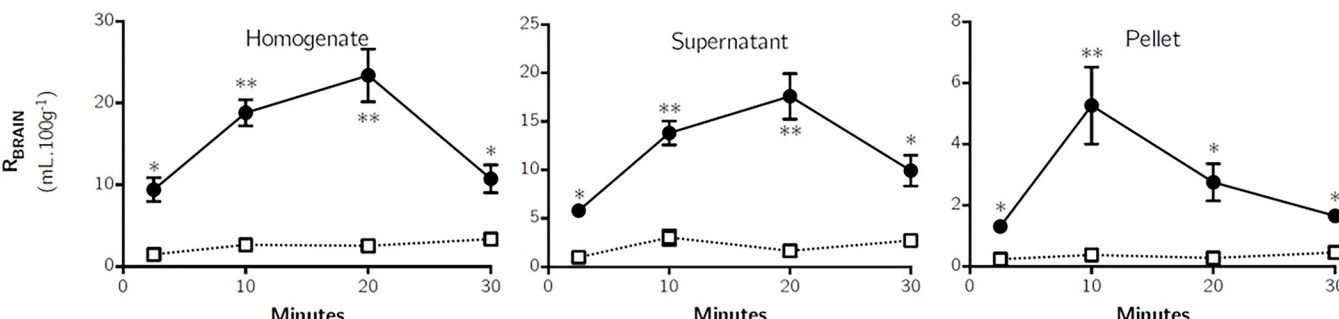

**Fig 6. Distribution of [³H]-ADMA and [¹⁴C]-sucrose in capillary depletion samples as a function of time.** Uptake is expressed as the percentage ratio of tissue to plasma (mL.100 g⁻¹). Each point represents the mean ± SEM of 5 animals. One-tailed, paired Student's t-tests comparing mean±SEM at each time point, *$p < 0.05$, **$p < 0.01$ (GraphPad Prism 6.0 for Mac).

A time-dependent increase in the distribution of [³H]-ADMA (corrected for [¹⁴C]sucrose) was observed in all brain regions up to 20 minutes (*e.g.* 9.86±1.43% after 2.5 minutes to 27.64 ±4.30% after 20 minutes in the frontal cortex), however this was followed by a decrease in the distribution of [³H]-ADMA at 30 minutes in all brain regions (*e.g.* 10.41±2.77% in the frontal cortex). No regional differences were observed ($p > 0.05$, two-way ANOVA

with Tukey's multiple comparison test for each brain region compared to all others within each time point). Due to a departure of linearity of the experimental points, multiple time uptake analysis (i.e. Eq 3) could not be used to calculate a $K_{in}$ for [³H]-ADMA, however, single time uptake analysis at 10 minutes was acceptable and revealed a $K_{in}$ of 2.49±0.35 µl.min⁻¹.g⁻¹ for [³H]-ADMA distribution into the frontal cortex (Table 2). There was no significant difference of the $K_{in}$ obtained from the eight brain regions sampled (one-way ANOVA with Dunnett's multiple comparisons of means, $p > 0.05$).

Fig 6 confirms the presence of [³H]-ADMA in the brain homogenate, the parenchyma-containing supernatant and endothelial cell enriched pellet samples at levels significantly higher than [¹⁴C]sucrose (one-tailed, paired Student's t-tests). This suggests that [³H]-ADMA can cross the BBB and enter the brain tissue. A very similar pattern of a time-dependent peak in the distribution of [³H]-ADMA after perfusing for 20 minutes was observed in whole brain homogenate and brain parenchyma (supernatant) following capillary depletion (Fig 6) to that observed in the brain regions (Fig 5). Interestingly, the peak in [³H]-ADMA distribution in the endothelial cell-enriched pellet following capillary depletion was observed earlier, after only 10 minutes of perfusion.

**CSF.**  Fig 7 shows that [³H]-ADMA could be detected in the CSF at 2.5 minutes being approximately 4%. Interestingly, the CSF distribution of [³H]-ADMA was like [¹⁴C]-sucrose at 2.5 and 30 minutes, but significantly higher at 10 and 20 minutes ($p<0.05$: One-tailed, paired Student's t-test comparing mean±SEM at each time point). The $K_{in}$ value determined by single-time uptake analysis at 10 minutes for [³H]-ADMA in the CSF was not significantly different to that in any of the brain regions samples (Table 2; one-way ANOVA with Dunnett's multiple comparisons of means, $p>0.05$).

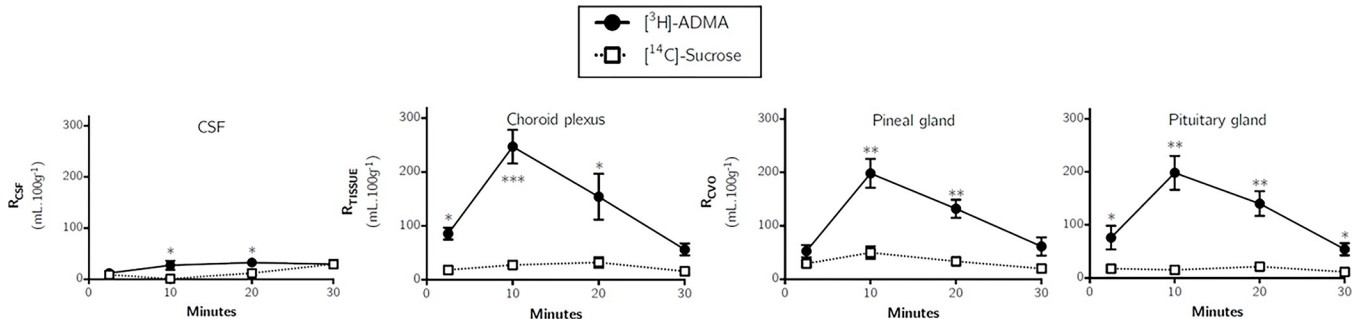

**Fig 7. Distribution of [³H]-ADMA and [¹⁴C]-sucrose in the CSF, pineal gland, choroid plexus and pituitary gland following *in situ* brain perfusion as a function of time.** Uptake is expressed as the percentage ratio of tissue or CSF to plasma (mL.100 g⁻¹). Each point represents the mean±SEM of 5 animals. One-tailed, paired Student's t-tests comparing mean±SEM at each time point, *$p < 0.05$, ** $p < 0.01$, *** $p < 0.001$ (GraphPad Prism 6.0 for Mac).

### Circumventricular organs

Fig 7 illustrates the distribution of [¹⁴C]-sucrose and [³H]-ADMA into the choroid plexus. The [¹⁴C]-sucrose distribution represents the extracellular space in this highly vascularised tissue. [³H]-ADMA accumulation was significantly greater than [¹⁴C]-sucrose accumulation at 2.5, 10 and 20 minutes. Importantly, the distribution of [³H]-ADMA into the choroid plexus reached a peak at 10 minutes of approximately 240% and then started to decrease until it reached approximately 40% at 30 minutes. A similar time-dependent increase and then decrease in the distribution of [³H]-ADMA was also observed in the pituitary gland and pineal gland (Fig 7). The mean [³H]-ADMA distribution in the choroid plexus, pituitary gland and pineal gland was not significantly different at any time point (uncorrected values, two-way ANOVA with Tukey's multiple comparison test for each sample compared to all others within each time point). Single-time uptake analysis could not be used to determine a $K_{in}$ for [³H]-ADMA transfer into the choroid plexus, pituitary gland and pineal gland as the [³H]-ADMA concentration in these tissues was higher than the concentration in the plasma.

### Comparison of [³H]-arginine and [³H]-ADMA

**CNS distribution.** The uptake of [³H]-arginine, [³H]-ADMA and [¹⁴C]-sucrose as a function of time is compared in S1–S3 Figs. The uptake of [³H]-arginine into the eight brain regions was significantly higher than that of [³H]-ADMA ($p < 0.05$) at all time points (Student's unpaired, two-tailed t-tests were used for the comparison of the two means) (S1 Fig). The rate of uptake of [³H]-ADMA ranged from 31.7–46.1% of the rate of uptake of [³H]-arginine into the different brain regions at 10 minutes (Table 2).

In the whole brain homogenate, and resulting supernatant following capillary depletion, [³H]-arginine distribution was significantly higher than [³H]-ADMA at 10 minute (p<0.05), 20 minute (p<0.05) and 30 minute (p<0.05) time points (Student's unpaired, two-tailed t-tests were used for the comparison of the two means) (S2 Fig). In the endothelial cell-enriched pellet resulting from capillary depletion, [³H]-arginine distribution was only significantly higher than [³H]-ADMA at the 20 minute (p<0.01) and 30 minute (p<0.01) time points. The rate of uptake of [³H]-ADMA ranged from 11.4–36.6% of the rate of uptake of [³H]-arginine into the different samples at 10 minutes (Table 2).

In the case of the CSF, pituitary gland, pineal gland and choroid plexus, [³H]-arginine distribution was significantly higher than [³H]- ADMA at the 20 minute (p<0.05) and 30 minute (p<0.05) time points only (Student's unpaired, two-tailed t-tests were used for the comparison

of the two means) (S3 Fig). The rate of uptake of [$^3$H]-ADMA ranged from 90.4–204.0% of the rate of uptake of [$^3$H]-arginine into the different samples at 10 minutes (Table 2).

**Self-inhibition experiments.** Fig 8 shows the effect of 100μM unlabelled arginine on the uptake of [$^3$H]-arginine into all brain regions after a 10-minute perfusion. These data indicate that [$^3$H]-arginine uptake into all eight brain regions is markedly self-inhibited by an average of approximately 67% (unpaired, one-tailed Student's t-test comparing means). S4 Fig shows the effect of 100μM unlabelled arginine on the distribution of [$^3$H]-arginine into capillary depletion samples. These data indicate that [$^3$H]-arginine uptake is inhibited by an average of approximately 56% (unpaired, one-tailed Student's t-test comparing means). S5 Fig also shows the effect of 100μM on the unlabelled *L*-arginine on the distribution of [$^3$H]-arginine into the CSF. These data indicate that [$^3$H]-arginine uptake into the CSF is inhibited by 62.2% ($p < 0.01$, unpaired, one-tailed Student's t-test comparing means). S5 Fig also shows the inhibitory effect of an excess of unlabelled arginine on the distribution of [$^3$H]-arginine into choroid plexus (57.3%; $p<0.001$), pineal gland (39.4%; $p<0.05$) and pituitary gland (48.1%; $p<0.05$) (unpaired, one-tailed Student's t-test comparing the means).

The uptake of [$^3$H]-ADMA is also significantly self-inhibited by 100 μM unlabelled ADMA in all brain regions after a 10-minute perfusion (Fig 9). The uptake of [$^3$H]-ADMA being significantly decreased by 60.3 to 74.3% when unlabelled ADMA was present. The same phenomenon was observed in capillary depletion samples (reduction 77.1 to 95.2%), CSF (reduction 89.1% although this did not attain statistical significance as only n of 2) and circumventricular organs (reduction 68.0–82.4%) (S6 and S7 Figs).

**Cross-inhibition experiments.** To determine the effect of ADMA on the transport of [$^3$H]-*L*-arginine across the blood-brain and blood-CSF barriers, [$^3$H]-arginine was co-perfused with different concentrations of unlabelled ADMA. Concentrations of ADMA were specifically selected to mimic plasma concentrations under normal conditions (0.5 μM), pathophysiological conditions (3.0 μM), and then supraphysiological concentrations (10, 100 and 500 μM). Fig 10 shows the effect of an excess of unlabelled ADMA on the uptake of [$^3$H]-arginine into all brain regions. These data indicate that [$^3$H]-arginine uptake is only significantly

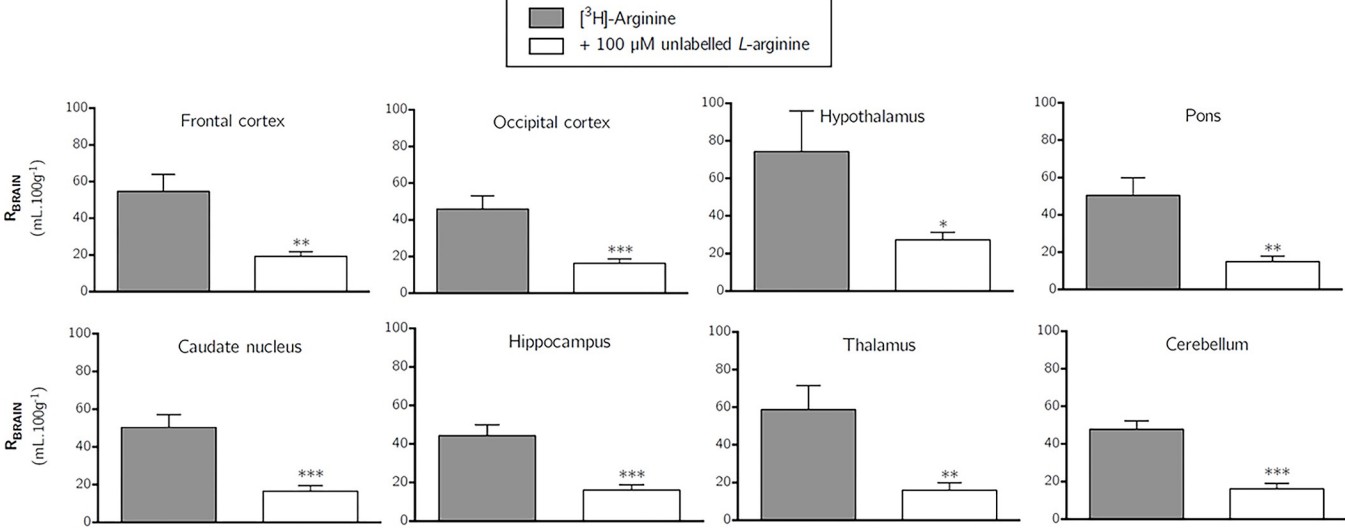

**Fig 8. The effect of 100μM unlabelled *L*-arginine on the uptake of [$^3$H]-arginine in the brain.** Uptake is expressed as the percentage ratio of tissue to plasma (mL.100 g$^{-1}$) and is corrected for [$^{14}$C]-sucrose (vascular space). Perfusion time is 10 minutes. Each bar represents the mean ± SEM of 6–7 animals (GraphPad Prism 6.0 for Mac). One-tailed, unpaired Student's t-tests comparing means. *$p < 0.05$, **$p < 0.01$, ***$p < 0.001$.

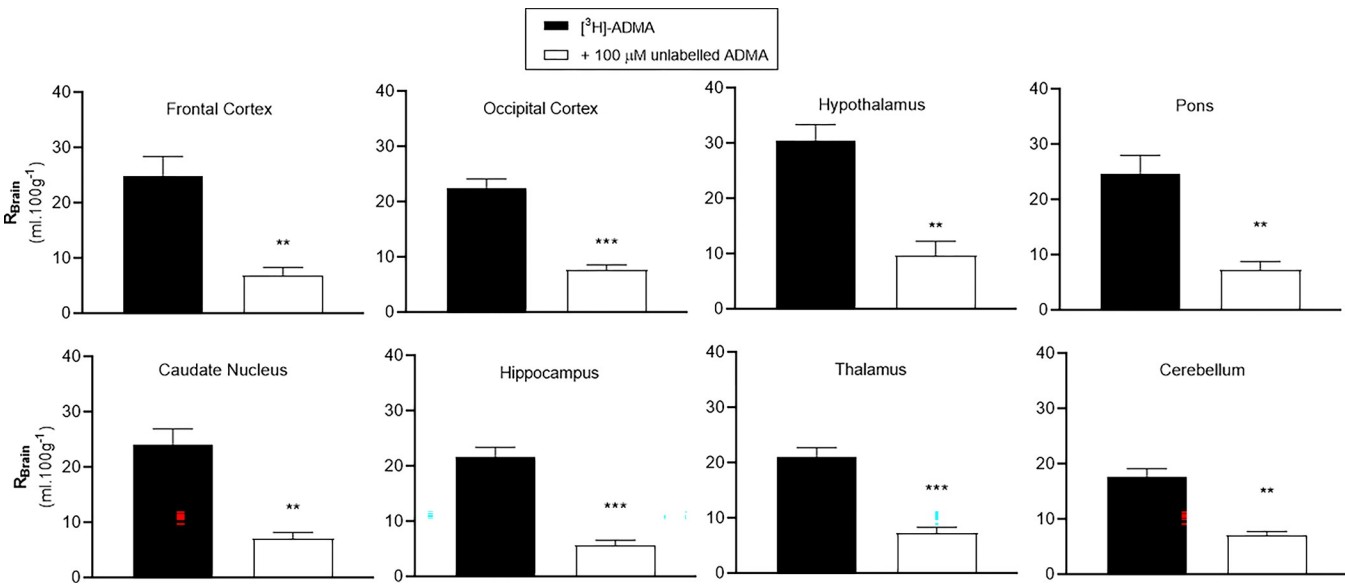

**Fig 9. The effect of 100μM unlabelled ADMA on the uptake of [³H]-ADMA in the brain.** Uptake is expressed as the percentage ratio of tissue to plasma (mL.100 g⁻¹) and is corrected for [¹⁴C]-sucrose (vascular space). Perfusion time is 10 minutes. Each bar represents the mean ± SEM of 4–5 animals (GraphPad Prism 6.0 for Mac). Unpaired Student's t-tests comparing means. *$p < 0.05$, **$p < 0.01$, ***$p < 0.001$.

inhibited by the highest concentration of unlabelled ADMA, which was 500 μM. [³H]-arginine uptake being significantly inhibited by approximately 70% into each of the brain regions (one-way ANOVA with Dunnett's post-hoc test comparing means to control, $p < 0.05$). Capillary depletion analysis of remaining whole brain tissue revealed that distribution of [³H]-arginine

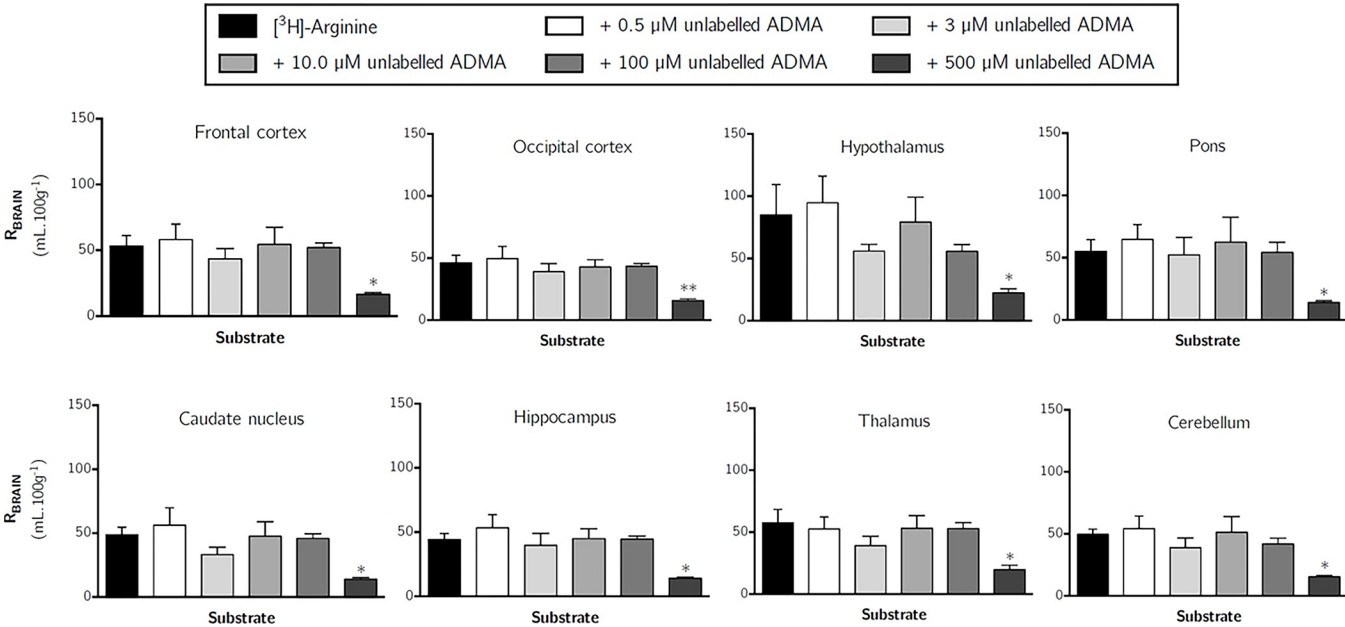

**Fig 10. The effect of unlabelled ADMA on the regional brain uptake of [³H]-arginine (10 minute perfusion).** Uptake is expressed as the percentage ratio of tissue to plasma (mL.100 g⁻¹) and is corrected for [¹⁴C]-sucrose (vascular space). Perfusion time is 10 minutes. Each bar represents the mean ± SEM of 4–7 animals. Asterisks represent one-way ANOVA with Dunnett's post-hoc tests comparing mean±SEM to control within each region/sample, *$p < 0.05$, **$p < 0.01$ (GraphPad Prism 6.0 for Mac).

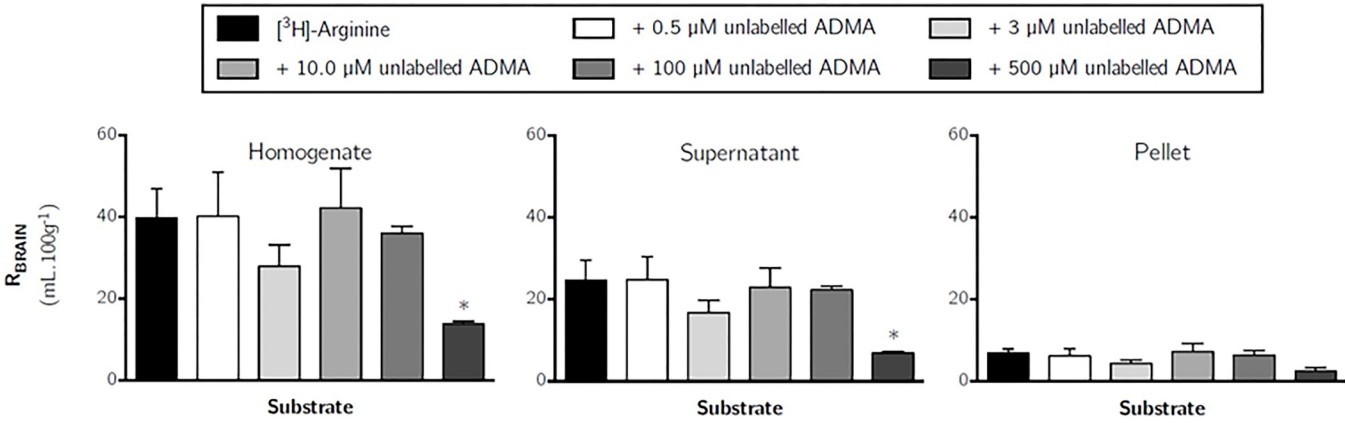

**Fig 11. The effect of unlabelled ADMA on the distribution of [³H]-arginine in capillary depletion samples (10 minute perfusion).** Uptake is expressed as the percentage ratio of tissue to plasma (mL.100 g⁻¹) and is corrected for [¹⁴C]-sucrose (vascular space). Perfusion time is 10 minutes. Each bar represents the mean ± SEM of 4–7 animals. Asterisks represent one-way ANOVA with Dunnett's post-hoc tests comparing mean±SEM to control within each region/sample, *$p < 0.05$ (GraphPad Prism 6.0 for Mac).

in only the whole brain homogenate and resulting supernatant (brain parenchyma) was inhibited by 500 μM unlabelled ADMA by 65.1% and 72.0%, respectively (Fig 11, $p < 0.05$). However, the distribution of [³H]-arginine into the cerebral capillary endothelial cells (i.e. pellet) was not affected by unlabelled ADMA even at the highest concentration of 500 μM (Fig 11). The distribution of [³H]-arginine in the CSF, pineal gland and pituitary gland was also not affected by any of the concentrations of unlabelled ADMA included in artificial plasma (Fig 12). However, 500 μM unlabelled ADMA inhibited the distribution of [³H]-arginine in the choroid plexus by 72.6% ($p < 0.01$).

Whilst 100 μM unlabelled ADMA did not inhibit the distribution of [³H]-arginine in any of the samples analysed (Figs 10–12), 100 μM *L*-arginine was sufficient to significantly inhibit distribution of [³H]-ADMA (S8–S10 Figs). S8 Fig shows the effect of an excess of unlabelled ADMA on the uptake of [³H]-arginine into all brain regions. These data indicate that [³H]-ADMA uptake is inhibited by 100 μM unlabelled *L*-arginine by up to 80.4% (one-tailed unpaired Student's t-test comparing means, p < 0.001). This trend was also mirrored in samples from capillary depletion analysis where 100 μM *L*-arginine inhibited [³H]-ADMA distribution in all samples by up to 80.8% (Unpaired, one-tailed Student's t-test was used to compare two means; S9 Fig).

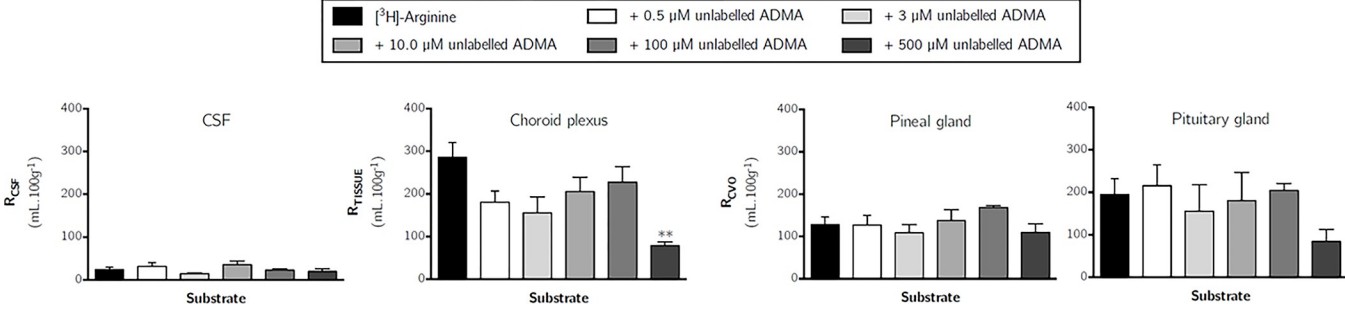

**Fig 12. The effect of unlabelled ADMA on the distribution of [³H]-arginine in CSF, choroid plexus and CVOs (10 minute perfusion).** Uptake is expressed as the percentage ratio of tissue or CSF to plasma (mL.100 g⁻¹). Perfusion time is 10 minutes. Each bar represents the mean ± SEM of 4–7 animals. Asterisks represent one-way ANOVA with Dunnett's post-hoc tests comparing mean±SEM to control within each region/sample, **$p < 0.01$ (GraphPad Prism 6.0 for Mac).

S10 Fig shows the effect of an excess of unlabelled *L*-arginine on the distribution of [³H]-ADMA into the CSF, pineal gland, choroid plexus and pituitary gland. These data indicate that [³H]-ADMA uptake is markedly inhibited by up to 78.6% (unpaired, one-tailed Student's t-test comparing means). The inclusion of unlabelled *L*-arginine however had no effect on the distribution of [³H]-ADMA in the CSF.

There was a difference in the degree of inhibition observed when [³H]-arginine distribution into the CNS was examined in the presence of 100 μM unlabelled arginine (significantly reduced) or 100 μM unlabelled ADMA (no change) (S11 Fig). However, there was no significant difference in the degree of inhibition observed on [³H]-ADMA distribution exerted by the inclusion of either 100 μM unlabelled ADMA (self-inhibition) or 100 μM unlabelled *L*-arginine (cross-competition (S11 Fig; p>0.05).

## Discussion

Circulating levels of the cationic amino acid, *L*-arginine, and its endogenously produced homologue, ADMA, are in a delicately poised equilibrium. As arginine is a NOS substrate and ADMA is a NOS inhibitor, dysregulation of this balance affects NO production and has well-documented implications in cardiovascular disease and brain conditions. In fact, impaired NO bioavailability represents the central feature of endothelial dysfunction, which is found in many diseases for example Alzheimer's disease [3]. The principal aim of our study is to explore ADMA and *L*-arginine transport at the blood-brain and blood-CSF barriers and discuss how it could be manipulated by arginine supplementation to treat NO dysregulation.

The first part of our study utilized a specialist chemical property database to explore the physicochemical characteristics of *L*-arginine and ADMA [21]. This database stated that *L*-arginine and ADMA had a molecular weight of 174.2 and 202.26 g/mol respectively and had a gross charge at physiological pH of +0.981. Interestingly, both *L*-arginine and ADMA exist as two microspecies at physiological pH. The major microspecies of both amino acids is a cationic (tripolar) amino acid (~98.1%) and the other microspecies is a zwitterion (dipolar) amino acid (~1.8%) (Fig 1). *L*-arginine had a lower lipophilicity than ADMA (predicted log D at pH 7.4 of -4.77 versus -3.99). This relationship was confirmed using octanol:saline partition coefficient measures.

The second part of our study compared the ability of [³H]-arginine and [³H]-ADMA to cross the mouse blood-CNS barriers and accumulate within the CNS. The integrity of the blood-CNS barriers was confirmed in all experiments by means of the marker molecule, [¹⁴C]-sucrose.

Multiple-time uptake studies detected [³H]-arginine in all brain regions (including frontal cortex, caudate nucleus, occipital cortex, hippocampus, hypothalamus, thalamus, pons and cerebellum) and CVOs (including the choroid plexus, pituitary gland and pineal gland) at significantly higher concentrations than [¹⁴C]-sucrose. The uptake of [³H]-arginine into the brain homogenate, brain tissue containing supernatant and capillary endothelial cell enriched pellet and the CSF was also significantly higher than [¹⁴C]- sucrose at all time points, except the pellet at 2.5 minutes where there was no difference. These results would indicate that [³H]-arginine can cross the cerebral capillary endothelium (i.e. the site of the BBB) to reach brain cells and that [³H]-arginine can cross the blood-CSF barrier (at the choroid plexuses) to reach the CSF.

While *in situ* brain/choroid plexus perfusion has previously been used to study [³H]-arginine uptake into the rat brain [32], no absolute values were published, and thus there are no published values to compare to the results described in our study. In our study, after 30 minutes, brain uptake of [³H]-arginine as a percentage of perfused plasma concentration reached

approximately 185.1% in the pons, and was similarly high in the majority of other brain regions, including the frontal cortex, caudate nucleus, occipital cortex, hippocampus and cerebellum (Fig 2). Thus after 30 minutes it appears that [³H]-arginine is sequestered within the brain parenchyma at levels greater than the artificial plasma levels ($R_{Brain} > 100\%$; [³H]-arginine artificial plasma concentration of 11.6 nM). One may predict that this would create a concentration gradient favouring flux of [³H]-arginine from the brain back into the plasma, at least under the experimental conditions presented here. However, this is unlikely to occur *in vivo* under normal conditions as the concentration of *L*-arginine in plasma is typically 100 μM in humans [33] and 140 μM in mice [34], while brain concentrations in both humans and mice are typically much lower at approximately 0.2–0.3 μM [35, 36]. This would create a concentration gradient that overwhelmingly favours diffusion and facilitated transport of *L*-arginine from plasma into the brain.

Multiple-time studies revealed that [³H]-ADMA uptake into all brain regions, capillary depletion samples (homogenate, supernatant and pellet) and pituitary gland were significantly higher than [¹⁴C]-sucrose at all time points (Figs 5–7). The uptake of [³H]-ADMA into the choroid plexus was also significantly higher than [¹⁴C]- sucrose at all time points except 30 minutes where there was no difference. The uptake of [³H]-ADMA into the pineal gland and the CSF was also significantly higher than [¹⁴C]-sucrose at all time points except 2.5 and 30 minutes where there was also no difference. These results would indicate that [³H]-ADMA can cross the cerebral capillary endothelium (i.e. the site of the BBB) to reach brain cells and that [³H]-ADMA can cross the blood-CSF barrier at the choroid plexuses to reach the CSF.

Another important point is that transport of [³H]-ADMA is bi-phasic in nature, with a peak in accumulation being reached at 10 to 20 minutes before decreasing at 30 minutes in all samples. It is possible that this decrease is related to the inhibition of NO production by ADMA, which would cause the cerebral blood vessels to vasoconstrict, reducing blood flow. This would decrease [³H]-ADMA delivery to the CNS. However, the [14C]-sucrose cerebrovascular space measured in the [3H]-ADMA study (Figs 5–7) was not statistically different to that measured in the [3H]-arginine study (Figs 2–4) so the inhibition of eNOS by ADMA is unlikely to generate the results observed. As the transport of the radiolabelled solute of interest back from the brain to the blood can be detected by a loss of linearity of the experimental points over time [26], the results of this present study indicate a significant CNS-to-blood efflux of [³H]-ADMA. The concentration of [³H]-ADMA achieved in the CNS samples (except the choroid plexus, pineal and pituitary glands) was lower than that in the artificial plasma at all time points. This would indicate that transporters (and not just passive diffusion) were involved in the brain-to-plasma efflux of [³H]-ADMA. However, the CSF (choroid plexus)-to-blood transfer could involve transporters and/or just be passive diffusion of [³H]-ADMA.

Interestingly, it has been shown that ADMA produced by one cell (e.g. an endothelial cell) can inhibit NO synthesis in a neighbouring cell (e.g. a macrophage) [37]. Transporters which can remove ADMA from cells include system y⁺ and system y⁺,L [17, 18]. Both these transport systems are expressed at the BBB [32, 38–40] and the system y⁺ transporter protein, CAT1, is also expressed at the choroid plexus [32, 41]. These facts provide further evidence that ADMA could be transferred from the CNS to the blood across the BBB and blood-CSF barrier. It is possible that the activity of these removal mechanism(s) become apparent above a threshold ADMA concentration due to transporter saturation hence the biphasic nature of the data shown in Figs 5–7. This would suggest that multiple saturable transport mechanisms with different kinetics are involved in ADMA transfer across cell membranes and/or that the transporter is bidirectional and is capable of transferring ADMA into and out of the cell with differing affinities. Interestingly, system y⁺ is a bidirectional uniporter and can transfer positively charged amino acids with a $K_m$ for influx being lower than the $K_m$ for efflux [42]. The

results so far described would indicate that [³H]-ADMA can cross the cerebral capillary endothelium (i.e. the site of the BBB) and the blood-CSF barrier in both directions (i.e. blood-to-CNS and CNS-to-blood) and that transporters are involved, at least in the brain-to-blood transfer.

During the experimental period when unidirectional transfer constants ($K_{in}$) can be calculated from multiple time uptake data using Eq 3, the amount of test substance in $V_j$ is roughly proportional to $C_{pl}$ and the test substance moves unidirectionally from plasma into brain tissue [31]. Consequently, multiple-time uptake analysis could not be used to calculate kinetic constants for either [³H]-arginine, as the former condition was not met, nor for [³H]-ADMA, as the latter condition was not met. Therefore, to calculate a transfer constant, studies at a perfusion time of 10 minutes were followed by single time uptake analysis (Table 2). Single-time uptake analysis to calculate a transfer constant using Eq 2 can be applied if entry of the test solute into the CNS is proportional to its plasma concentration, the concentration in the CNS is less than the concentration in the plasma and efflux (CNS-to-blood) is much smaller than influx (blood-to-CNS) of the test solute and therefore can be ignored [27]. Therefore, a transfer constant could be calculated for [³H]-arginine and [³H]-ADMA distribution into most CNS regions at a perfusion time of 10 minutes (Table 2). Although, it was not possible to determine a transfer constant by single-time uptake analysis for [³H]-arginine or [³H]-ADMA distribution into the choroid plexus, pineal gland and pituitary gland, as its concentration in the CNS was more than the plasma (Figs 4 and 7).

Of the two radiolabelled cationic amino acids studied, [³H]-arginine demonstrated the greatest ability (more than two-fold in most cases) to cross the BBB *in situ* and accumulate within all eight brain regions and the capillary depletion samples (S1 and S2 Figs; Table 2). We have previously observed a greater ability (more than -three to -five-fold) of [³H]-arginine (7 nM) when compared to [³H]-ADMA (38 nM) to cross brain capillary endothelial cell membranes by means of accumulation studies in hCMEC/D3 cells [12].

Both *L*-arginine and ADMA have the same gross charge of +0.981 and a major microspecies at physiological pH which has a single positive charge (Fig 1). As arginine has a higher hydrophilicity when compared to ADMA and is of similar hydrophilicity to sucrose, the greater ability of arginine to accumulate within all regions of the brain *in situ* may be related to the use of specific transporters as also suggested by our previous *in vitro* study [12]. These transporters may aid the movement of [³H]-arginine into the brain and / or aid the removal of [³H]-ADMA out of the CNS.

Our self-inhibition studies using the *in situ* brain/choroid plexus perfusion technique and 100 μM unlabelled arginine at 10 minutes caused a reduction in the uptake of [³H]-arginine (11.6 nM) by approximately 39.4–73.0% into all the CNS samples measured. This would indicate the use of transporters to transfer [³H]-arginine across the mouse blood-brain and blood-CSF barriers. The results of our study would also confirm that there are (at the very least) transporters for [³H]-arginine on the luminal membrane of the cerebral capillary endothelium (pellet sample) and the blood-side of the choroid plexus.

Saturable transport of radiolabelled *L*-arginine has previously been observed across the rat BBB using the *in situ* brain perfusion technique [32, 43] and at the blood-CSF barrier using the isolated perfused sheep choroid plexus technique [44, 45]. Although we did not determine the half-saturation constant ($K_m$) of [³H]-arginine transport across the blood-CNS interfaces in our mouse study, our data would align with the half-saturation constant ($K_m$) for arginine at the BBB previously determined using a similar method in rats (56 ± 9 μM) and the brain uptake index method in rats (40 ± 24 μM (±standard deviation)) [43] and the blood-side (basolateral membrane) of the choroid plexus epithelium determined using the isolated perfused choroid plexus of the sheep (25.4 ± 5.1 μM) [32, 44]. Kinetic studies in different types of

human and animal cells have indicated that sodium-independent influx of arginine at physiological concentrations occurs predominately by one saturable transport system and has extracellular $K_m$ values ranging from 25 to 200 μM [42, 46–48].

Interestingly, the cationic amino acids transport system, system-$y^+$, has been detected at both luminal and abluminal membranes in brain capillary endothelial cells, with prevalence at the abluminal membrane [40]. In addition, the half-saturation constants ($K_m$) for arginine transport by murine CAT1 expressed in Xenopus oocytes and the $y^+$-system expressed in human fibroblasts is 77 ± 2 μM and 40 ± 0.05 μM, respectively [32, 48, 49]. It is therefore plausible that the transporters for arginine detected at the blood-CNS interfaces in our study are system-$y^+$ transporters. We have previously identified the presence of the $y^+$-system protein, CAT1, in human BBB cells (hCMEC/D3) using Western blotting and immunofluorescence [12, 50] and CAT 1 is expressed at the blood-CSF barrier specifically the rat choroid plexus [32, 41]. Furthermore, the human gene CAT-1 shares 87.6% amino acid identity with the mouse gene [51]. Importantly, the high degree of sequence conservation has been shown to reflect functional similarity of the human transporters with their mouse homologues [52, 53]. *L*-arginine uptake by oocytes expressing hCAT-1 or mCAT-1 having a $K_m$ of 110–160 μM and 140–250 μM respectively [52–55]. This would further suggest that there would be a functional similarity of the human and mouse CATs.

Saturable transport of [$^3$H]-ADMA (38 nM-3 μM) into cultured human cerebral capillary endothelial cells has previously been demonstrated by our group and was suggested to be due to system $y^+$ -activity [12]. In our present *in situ* mouse brain perfusion study, self-inhibition studies using 100 μM unlabelled ADMA at 10 minutes reduced the uptake of [$^3$H]-ADMA (62.5 nM) by approximately 60.3–95.2% into all the CNS samples measured. This suggests the use of transporters to transfer [$^3$H]-ADMA across both mouse blood-brain and blood-CSF barriers. Importantly, this is the first study to reveal saturable transport of ADMA at the blood-CSF barrier (choroid plexus). It remains to be seen if these transporters are saturated by ADMA at physiological plasma concentrations which are 0.30–1.57 μM in humans [54] and 1.07–1.58 μM in mice [55].

The decrease in the concentration of [$^3$H]-arginine and [$^3$H]-ADMA in the presence of unlabelled arginine or ADMA is consistent with inhibition of transporters involved in uptake but it could also be due to stimulation of cellular efflux. As a classical system $y^+$ transporter, CAT-1, is bi-directional in its transport activity and can result in the exchange of cationic amino acids between the two sides of the membrane [56, 57]. In fact, transport is faster when substrate is present at the opposite (*trans*-) side of the membrane [57]. This effect is exhibited by system-$y^+$ and called trans-stimulation [11].

Interestingly, in our *in situ* brain perfusion study, transport of [$^3$H]-arginine (11.6 nM) across the blood-CNS barriers was insensitive to inhibition by 0.5–100 μM unlabelled ADMA, but sensitive to inhibition by 500 μM unlabelled ADMA. In contrast, the transport of [$^3$H]-ADMA (62.5 nM) into all samples was sensitive to inhibition by 100 μM arginine (inhibition ranging from 64.3–80.8%). This is in agreement with studies using human embryonic kidney cells (HEK293) stably overexpressing CAT1 and vector-transfected control cells, which have shown that 100 μM ADMA inhibition of CAT1-mediated transport of *L*-arginine was undetectable (CAT1 having an $IC_{50}$ 758 (460–1251) μM), but 100 μM L-arginine inhibition of CAT1-mediated cellular uptake of ADMA was detectable (CAT1 having an $IC_{50}$ 227 (69–742) μM) [18]. This suggests that higher concentrations of unlabelled ADMA (i.e. >100 μM) would be needed to significantly affect [$^3$H]-arginine transport by CAT-1 in agreement with our study. In contrast, another study has revealed that 100 μM unlabelled ADMA can exert a significant inhibitory effect on [$^3$H]-arginine transport, likely via CAT, into human dermal

microvascular endothelial cells [58]. Although, as also observed in our study, lower concentrations of ADMA (2.5–10 μM) did not affect [³H]-arginine transport in these cells.

In contrast to these cross-inhibition studies, the transport of a cationic amino acid analogue, [³H]-eflornithine (720 nM), into hCMEC/D3 cells appears much more sensitive to inhibition by 100 μM ADMA than 100 μM *L*-arginine [50].

Interestingly, cross-competition studies revealed that [³H]-arginine uptake into the isolated incubated rat choroid plexus can be inhibited by another methylated arginine, $N^G$-methyl-*L*-arginine, at a concentration of 500 μM [59]. This *in vitro* method focuses on molecule movement across the apical/CSF side of the choroid plexus, in contrast to the luminal/blood side of the choroid plexus that is examined by the *in situ* brain/choroid plexus perfusion technique.

Interactions between arginine and ADMA have previously been demonstrated in relation to cellular transport [11]. There is evidence for both inhibition of uptake and increased efflux of ADMA in the presence of arginine [11, 18]. For example, Shin et al. in 2017 demonstrated an enhanced efflux of endogenous arginine and ADMA from a human umbilical vein endothelial cell line due to extracellular exogenous arginine exposure [11]. This is consistent with the trans-stimulation of system $y^+$ transporters.

Our study used the *in situ* brain/choroid plexus perfusion method in adult BALB/c mice to study the transport of [³H]-arginine and [³H]-ADMA across the BBB and blood-CSF barrier. Importantly, these barriers are very complex interfaces and cannot be replicated fully using cell culture methods. However, the *in situ* results are difficult to interpret conclusively due to multiple interacting factors such as: i) transport of the test molecule may occur into and out of the CNS (as suggested by the ADMA multiple time uptake data presented) ii) transfer of test molecules from brain tissue to CSF and CSF to brain tissue iii) loss of integrity of the radiolabels to the test molecule within the brain tissue and CSF and then removal of the radiolabel from the CNS iv) different transporters for the test molecule may be expressed at the BBB and blood-CSF barrier. Despite these limitations, consideration of all these facts still allows important conclusions to be drawn from the results. In addition, the data obtained from these *in situ* studies can be compared to data obtained *in vitro* as a means of further validating the methods and data. The characteristics of mouse models are of interest due to the availability of transgenic animals, oligoprobes, antibodies and mouse models of disease. Thus, expansion of our knowledge of mouse cells is of relevance to scientific investigations. Furthermore, mice are ideal for these studies as their BBB is similar in structure and function to that of humans.

Our studies confirm that the transport systems used by ADMA and *L*-arginine at the blood-CNS barriers appear to be shared to some degree, as they each affect the others' transport (Figs 10–12; S8–S10 Figs) and the transport of other cationic amino acids [50], but to a differing extent (S11 Fig). Understanding this relationship is of clinical relevance as dietary supplementation with *L*-arginine has been shown to alleviate endothelial dysfunctions caused by impaired NO synthesis [13, 14]. Oral supplementation with one dose of 10g arginine has been shown to produce a peak arginine plasma concentration of 200–300 μM [60]. This present study would support the theory that arginine supplementation can increase NO production not by inhibiting arginine influx, but by inhibition of [³H]-ADMA influx and/ or stimulating the efflux of [³H]-ADMA. This would ultimately lower the intracellular concentration of ADMA, but not affect the intracellular concentration of *L*-arginine significantly. The reasons for this are as follows. Firstly, our data supports the observations that the transporter for arginine would be operating at nearly maximum capacity in other words is nearly fully saturated ($K_m$ 25–77 μM) within the physiological plasma concentration range of endogenous arginine (human ~100 μM and mouse ~140 μM) [34, 33]. This together with the facts that: (i) *L*-arginine transport by CAT-1 is a pre-requisite of NO production by the endothelium [61, 62] and ii) eNOS is normally already saturated ($K_m$ ~3–36 μM) with endogenous intracellular

*L*-arginine (840±90 μM) [15, 16, 62, 63], would indicate that raising the plasma concentration of arginine would not increase NO production by increasing the intracellular concentration of arginine [64]. Secondly, our data provides *in situ* evidence that ADMA can cross cell membranes using transporters, transport occurs in both directions and that arginine can reduce ADMA cellular accumulation. Thus, it is possible that supplementation with arginine would affect ADMA transport (efflux) resulting in the displacement of intracellular ADMA from eNOS (inhibition constant ($K_i$) 0.9μM) [16] and increased NO production [54, 64, 65]. Although a physiological intracellular concentration of ADMA of about 3.6 μM is thought to have only a modest effect (10%) on endothelial NO production [16]. Pathophysiological plasma concentrations of ADMA, which are 3- to 9-fold higher, inhibit NO production by 30–70% [16]. Studies in hypercholesterolemic young adults have demonstrated that arginine supplementation alleviates endothelial dysfunction [14]. Although ADMA was not measured in this clinical study, hypercholesterolemia is associated with elevated plasma ADMA concentrations (2.17±0.15 μM versus 1.03±0.09 μM) [66]. Overall, this suggests that arginine supplementation could affect ADMA transport (efflux) and consequently increase NO production particularly if ADMA plasma concentrations are higher than normal. Although it must be noted that other mechanisms may also affect intracellular ADMA concentrations and/or NO production. For example, arginine inhibits ADMA catabolism by competing with ADMA for dimethylarginine dimethylaminohydrolase (DDAH) [66] and ADMA may also increase the apparent $K_m$ of NOS for arginine [54, 65]. Indeed, arginine supplementation may cause multiple mechanisms to act in parallel. The relative importance of CAT, DDAH on intracellular ADMA concentrations and/or NO production depending on pathophysiological circumstance and ADMA plasma concentrations. This complex relationship may explain the absence of measurable clinical benefits of arginine supplementation in some clinical studies [5]. It also suggests that more detailed consideration be given to the ADMA plasma concentration of patients before recommending arginine supplementation as a therapeutic strategy. Higher plasma concentrations of ADMA indicating that arginine supplementation would be more successful at increasing endothelial NO production and leading to beneficial effects.

## Conclusion

This study has examined the transport of [$^3$H]-*L*-arginine and [$^3$H]-ADMA across BBB and blood-CSF barrier using physicochemical assessment and an *in situ* mouse brain and choroid plexus perfusion technique. These tripolar cationic amino acids are structurally related but have opposite functions. *L*-Arginine being the exclusive physiological substrate for the NOS family, which synthesizes NO in endothelial cells and neuronal cells, and ADMA being an inhibitor of NOS and so inhibiting NO production.

Results indicate that both [$^3$H]-arginine and [$^3$H]-ADMA have a gross charge at pH 7.4 of +0.981, but [$^3$H]-arginine has a lower lipophilicity than [$^3$H]-ADMA. Both [$^3$H]-arginine and [$^3$H]-ADMA can cross the blood-brain and blood-CSF barriers, but have differing abilities to accumulate in the brain and CSF. This is likely related to their differing ability to use specific transporters for cationic amino acids expressed at these interfaces. This is the first study to present (i) evidence of saturable [$^3$H]-ADMA transport at the blood-CSF barrier (choroid plexus) and (ii) that [$^3$H]-ADMA undergoes significant CNS-to-blood efflux across both blood-brain and blood-CSF barriers. Our results also indicate that there is at least some overlap in the specificity of the transporter(s) for arginine and ADMA at the blood-CNS interfaces. For example, unlabelled arginine decreased the CNS accumulation of [$^3$H]-ADMA. It is not yet known if this is due to inhibiting [$^3$H]-ADMA influx or stimulating [$^3$H]-ADMA efflux.

Our data also provides an explanation for the *L*-arginine paradox supporting the hypothesis that increased NO production due to arginine supplementation is likely related to enhanced ADMA efflux rather than increased arginine influx. This is of clinical relevance as NO is an endogenous endothelial vasodilator, and its enhanced production is beneficial in diseases that are associated with endothelial dysfunction.

In the future we hope to explore [³H]-arginine and [³H]-ADMA transport across the blood-brain and blood-CSF barriers in more detail determining the pharmacokinetic constants and the identity of the transporters involved. This may further contribute to understanding the contribution of ADMA efflux to any potential benefits of arginine supplementation.

## Supporting information

**S1 Fig. Comparative uptake of [³H]-arginine, [³H]-ADMA and [¹⁴C]-sucrose as a function of time measured by *in situ* brain perfusion in anaesthetized mice.** Uptake is expressed as the percentage ratio of tissue to plasma (mL.100 g$^{-1}$). Perfusion fluid contained either [³H]-arginine and [¹⁴C]-sucrose (open markers) or [³H]-ADMA and [¹⁴C]-sucrose (filled markers). Each point represents the mean ± SEM of 4–7 animals (GraphPad Prism 6.0 for Mac).
(PDF)

**S2 Fig. Distribution of [³H]-arginine, [³H]-ADMA and [¹⁴C]-sucrose in capillary depletion samples as a function of time.** Uptake is expressed as the percentage ratio of tissue to plasma (mL.100 g$^{-1}$). Each point represents the mean ± SEM of 5 animals. Kin and Vi values were determined as the slope and ordinate intercept of the computed regression lines (GraphPad Prism 6.0 for Mac).
(PDF)

**S3 Fig. Comparative distribution of [³H]-arginine, [³H]-ADMA and [¹⁴C]-sucrose in the CSF, pineal gland, choroid plexus and pituitary gland following *in* situ brain perfusion as a function of time.** Uptake is expressed as the percentage ratio of tissue or CSF to plasma (mL.100 g$^{-1}$). Each point represents the mean ± SEM of 4–7 animals (GraphPad Prism 6.0 for Mac).
(PDF)

**S4 Fig. The effect of 100μM un-labelled *L*-arginine on the distribution of [³H]-arginine in capillary depletion samples.** Uptake is expressed as the percentage ratio of tissue or CSF to plasma (mL.100 g-1). Brain samples have been corrected for [¹⁴C]-sucrose (vascular space). Perfusion time is 10 minutes. Each bar represents the mean ± SEM of 6–7 animals. (GraphPad Prism 6.0 for Mac). One-tailed unpaired Student's t-test comparing means. $^{*}$p < 0.05, $^{**}$p < 0.01.
(PDF)

**S5 Fig. The effect of 100μM un-labelled L-arginine on the distribution of [³H]-arginine in the CSF, choroid plexus and circumventricular organs.** Uptake is expressed as the percentage ratio of tissue or CSF to plasma (mL.100 g-1). Perfusion time is 10 minutes. Each bar represents the mean ± SEM of 6–7 animals (GraphPad Prism 6.0 for Mac). One-tailed unpaired Student's t-test comparing means. $^{*}p < 0.05$, $^{**}p < 0.01$, $^{***}p < 0.001$.
(PDF)

**S6 Fig. The effect of 100μM un-labelled ADMA on the uptake of [³H]-ADMA in the capillary depletion samples.** Uptake is expressed as the percentage ratio of tissue to plasma (mL.100 g$^{-1}$) and is corrected for [¹⁴C]-sucrose (vascular space). Perfusion time is 10 minutes.

Each bar represents the mean ± SEM of 4 animals (GraphPad Prism 6.0 for Mac). Unpaired Student's t-tests comparing means. $*p < 0.05$, $**p < 0.01$, $***p < 0.001$.
(PDF)

**S7 Fig. The effect of 100μM un-labelled ADMA on the uptake of [$^3$H]-ADMA in the CSF, choroid plexuses and circumventricular organs.** Uptake is expressed as the percentage ratio of tissue to plasma (mL.100 g$^{-1}$). Perfusion time is 10 minutes. Each bar represents the mean ± SEM of 2–5 animals (GraphPad Prism 6.0 for Mac) for each region in the control group. The unlabelled ADMA group had varying numbers of experiments for example, CSF (n = 2), pineal gland (n = 3), choroid plexus (n = 4) and pituitary gland (n = 4) (GraphPad Prism 6.0 for Mac). Unpaired Student's t-tests comparing means.$*p<0.05$, $**p<0.01$ and $***p < 0.001$.
(PDF)

**S8 Fig. The effect of unlabelled *L*-arginine on the regional brain uptake of [$^3$H]-ADMA (10 minute perfusion).** Uptake is expressed as the percentage ratio of tissue to plasma (mL.100 g$^{-1}$) and is corrected for [$^{14}$C]-sucrose (vascular space). Each bar represents the mean ± SEM of 5 animals (GraphPad Prism 6.0 for Mac). One-tailed unpaired Student's t-test comparing means $***p < 0.001$.
(PDF)

**S9 Fig. The effect of unlabelled *L*-arginine on the distribution of [$^3$H]-ADMA in capillary depletion samples (10 minute perfusion).** Uptake is expressed as the percentage ratio of tissue to plasma (mL.100 g$^{-1}$) and is corrected for [$^{14}$C]-sucrose (vascular space). Each bar represents the mean ± SEM of 5 animals (GraphPad Prism 6.0 for Mac). Unpaired, one-tailed Student's t-test was used to compare two means. $**p<0.01$ and $***p < 0.001$.
(PDF)

**S10 Fig. The effect of unlabelled *L*-arginine on the distribution of [$^3$H]-ADMA in CSF, choroid plexus and CVOs (10 minute perfusion).** Uptake is expressed as the percentage ratio of tissue or CSF to plasma (mL.100 g$^{-1}$). Each bar represents the mean ± SEM of 5 animals (GraphPad Prism 6.0 for Mac). Unpaired, one-tailed Student's t-test was used to compare two means. $*p<0.05$, $**p<0.01$ and $***p < 0.001$.
(PDF)

**S11 Fig. Effect of either 100 μM unlabelled ADMA or 100 μM unlabelled *L*-arginine on the respective uptake and distribution of [$^3$H]-ADMA or [$^3$H]-arginine in the frontal cortex and choroid plexus.** Uptake is expressed as the percentage ratio of tissue or CSF to plasma (mL.100 g$^{-1}$). Each bar represents the mean ± SEM of 4–5 animals (GraphPad Prism 6.0 for Mac). One-way ANOVA with Dunnett's post-hoc test comparing means to control ([$^3$H]-ADMA only), $***p < 0.001$).
(PDF)

**S1 Dataset. The octanol-saline partition coefficient values behind the means and standard errors of the means reported for [$^3$H]-arginine, [$^3$H]ADMA and [$^{14}$C]sucrose.**
(XLSX)

**S2 Dataset. The *in situ* brain perfusion data values behind the means and standard errors of the mean reported for [$^3$H]-arginine and [$^{14}$C]sucrose.**
(XLSX)

**S3 Dataset. The *in situ* brain perfusion data values behind the means and standard error of the mean reported for [³H]-ADMA and [¹⁴C]-sucrose.**
(XLSX)

## Acknowledgments

This paper includes data from the PhD thesis of Mehmet Fidanboylu [67]. Abstracts of this work have been published [68].

## Author Contributions

**Conceptualization:** Mehmet Fidanboylu, Sarah Ann Thomas.

**Data curation:** Mehmet Fidanboylu.

**Formal analysis:** Mehmet Fidanboylu, Sarah Ann Thomas.

**Funding acquisition:** Sarah Ann Thomas.

**Investigation:** Mehmet Fidanboylu, Sarah Ann Thomas.

**Project administration:** Mehmet Fidanboylu, Sarah Ann Thomas.

**Resources:** Sarah Ann Thomas.

**Supervision:** Sarah Ann Thomas.

**Validation:** Mehmet Fidanboylu.

**Visualization:** Mehmet Fidanboylu, Sarah Ann Thomas.

**Writing – original draft:** Mehmet Fidanboylu, Sarah Ann Thomas.

**Writing – review & editing:** Mehmet Fidanboylu, Sarah Ann Thomas.

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
