## [Decision Letter · Decision Letter 0]

1 Aug 2024

PONE-D-24-21535

L -Arginine and asymmetric dimethylarginine (ADMA) transport across the mouse blood-brain and blood-CSF barriers: evidence of saturable transport at both interfaces and CNS to blood efflux.

PLOS ONE

Dear Dr. Thomas,

Thank you for submitting your manuscript to PLOS ONE. After careful consideration, we feel that it has merit but does not fully meet PLOS ONE’s publication criteria as it currently stands. Therefore, we invite you to submit a revised version of the manuscript that addresses the points raised during the review process.

We look forward to receiving your revised manuscript.

Kind regards,

Emre Avci

Academic Editor

PLOS ONE

2. In order to comply with PLOS ONE's guidelines for non-human primate experiments (http://journals.plos.org/plosone/s/submission-guidelines#loc-non-human-primates), please provide additional details regarding housing conditions, feeding regimens, environmental enrichment, and all relevant steps taken to alleviate suffering (anesthesia, analgesia, details about humane endpoints, euthanasia, etc.). Also indicate how often animal care staff monitored the health and well-being of the animals and the criteria used to make such assessments. Lastly, specify the disposition of animals at the end of the study (e.g. euthanasia, returned to home colony, etc.). If animals were euthanized following the study, please provide the method of sacrifice.

Additional Editor Comments (if provided):

Reviewers' comments:

Reviewer's Responses to Questions

**Comments to the Author**

1. Is the manuscript technically sound, and do the data support the conclusions?

Reviewer #1: Yes

Reviewer #2: Yes

2. Has the statistical analysis been performed appropriately and rigorously? 

Reviewer #1: Yes

Reviewer #2: Yes

3. Have the authors made all data underlying the findings in their manuscript fully available?

Reviewer #1: Yes

Reviewer #2: Yes

4. Is the manuscript presented in an intelligible fashion and written in standard English?

Reviewer #1: Yes

Reviewer #2: Yes

5. Review Comments to the Author

Reviewer #1: The introduction is well-written, easy to understand, and effectively conveys the necessary information. However, here are a few minor suggestions to further enhance its quality:

- L65-70: Please cite necessary studies about this crucial topic.

The experimental design is well-planned and well-written. The control experiments and normalization of data is clearly explained. I recommend the authors to explain these points shortly to increase clarity:

- L127-147: Please provide references to support the advantages mentioned in the “in situ brain/choroid plexus perfusion” technique.

- L163: How the CSF sample was taken is not clearly explained.

- Could the authors explain why they chose to perform one-tailed student-t tests instead of two-tailed and two-tailed test for one analysis (L490)? What are the significant differences based on two-tailed test, are they the same or different from current differences? Two-tailed test might be more reliable in terms of significance validity.

In Discussion:

- L648-..: The authors mention that there might be a concentration gradient favouring flux of [3H]-arginine from the brain back into the plasma, however, the plasma concentrations of L-arginine is different in humans and mice. The adaptation of the results to humans could be different, how do the authors explain that [3H]-arginine sequestration to brain would be similar in humans in that case?

- The changes of arginine concentrations might result in changes in ADMA concentrations endogenously in brain, rather than transport mechanisms, as they regulate one another’s metabolism. The opposite is also possible. What would be the authors’ comments about this? (Please see: https://doi.org/10.1016/j.ymgme.2007.04.017)

Reviewer #2: The author has done tremendous research on the CNS delivery of L-arginine and ADMA. Overall, the manuscript sounds good. Please, see the mentioned below comments-

1. Mechanism of ADMA Transport: While the study provides valuable data on ADMA and L-arginine distribution, it would be strengthened by delving deeper into the specific mechanisms of ADMA transport across the blood-brain barrier. Are there specific transporters involved? Is there evidence of active transport or passive diffusion?

2. Specificity of Arginine Supplementation: The study suggests arginine supplementation as a potential therapeutic strategy. However, it would be beneficial to explore the specificity of this approach. Does arginine supplementation primarily increase L-arginine levels or also affect ADMA levels? Could this approach lead to unintended consequences?

3. Clinical Relevance: The study highlights the importance of ADMA and L-arginine in vascular health. It would be valuable to discuss the clinical implications of the findings. How could these findings be translated into potential diagnostic or therapeutic strategies for patients with vascular diseases?

4. Comparison to Other Studies: The authors could strengthen their discussion by comparing their findings to existing literature on ADMA and L-arginine transport in the brain. Are there any discrepancies or areas of agreement with previous studies?

5. Limitations of the Study: The authors should acknowledge the limitations of their study, such as the use of animal models and the potential for extrapolation to human physiology. This would enhance the transparency and credibility of their research.

6. Future Directions: The study concludes with a call for further research. The authors could expand on this by outlining specific future directions for their research, such as investigating the role of specific transporters, exploring the effects of different arginine supplementation strategies, or conducting clinical trials to assess the therapeutic potential of their findings.

7. Statistical Analysis: The authors should provide more details about the statistical analysis used in their study. This would enhance the rigor and reproducibility of their findings.

8. Data Availability: The authors should clearly state the availability of their data, including the raw data and any supporting information. This would promote transparency and allow other researchers to replicate and build upon their findings.

9. Ethical Considerations: The authors should address any ethical considerations related to their study, particularly if they involved animal models. This would demonstrate their commitment to responsible research practices.

10. Clarity and Conciseness: The authors could improve the clarity and conciseness of their writing. This would make their findings more accessible to a wider audience.

11. Please improve the picture images of all, as the image is not in the higher pixel, and the image is getting blurred.

6. PLOS authors have the option to publish the peer review history of their article (what does this mean?). If published, this will include your full peer review and any attached files.

---

## [Author Response · Author response to Decision Letter 0]

17 Sep 2024

Manuscript number: PONE-D-24-21535

L -Arginine and asymmetric dimethylarginine (ADMA) transport across the mouse blood-brain and blood-CSF barriers: evidence of saturable transport at both interfaces and CNS to blood efflux.

Dear Editors and Reviewers,

Thank you for taking the time to review this manuscript and provide constructive feedback.

Please find following our responses to the editors and reviewers comments:

EDITOR REQUIREMENTS 1

AUTHORS RESPONSE

We have formatted the manuscript to ensure it meets all of PLOS ONE’s style requirements. 

EDITOR REQUIREMENT 2

2. In order to comply with PLOS ONE's guidelines for non-human primate experiments (http://journals.plos.org/plosone/s/submission-guidelines#loc-non-human-primates), please provide additional details regarding housing conditions, feeding regimens, environmental enrichment, and all relevant steps taken to alleviate suffering (anesthesia, analgesia, details about humane endpoints, euthanasia, etc.). Also indicate how often animal care staff monitored the health and well-being of the animals and the criteria used to make such assessments. Lastly, specify the disposition of animals at the end of the study (e.g. euthanasia, returned to home colony, etc.). If animals were euthanized following the study, please provide the method of sacrifice.

AUTHORS RESPONSE

We did not use non-human primates in this study. However, we have provided additional details under the animals and anaesthesia section Lines 147-173.

EDITOR REQUIREMENT 3

AUTHORS RESPONSE:

We confirm we have reviewed the reference list. As far as we are aware no article that we cite in this paper has, as yet, been retracted.

REVIEWER 1:

1. REVIEWER 1 COMMENT: 

- L65-70: Please cite necessary studies about this crucial topic.

AUTHOR RESPONSE:

Original manuscript version: Lines 65-70 states this:

‘As L-arginine is the precursor for NO synthesis (7) and ADMA a potent endogenous inhibitor of NO synthesis (5) (6), the interplay between the transport of these two cationic amino acids at the blood-brain barrier (BBB) and blood-cerebrospinal fluid (CSF) barriers is likely to directly relate to NO production at these interfaces and within the central nervous system (CNS). In fact, the arginine/ADMA plasma concentration ratio is widely considered to be an important indicator of NO bioavailability’. 

REFERENCES

(5) Vallance P, Leone A, Calver A, Collier J, Moncada S. Accumulation of an endogenous inhibitor of nitric oxide synthesis in chronic renal failure. Lancet. 1992 Mar 7;339(8793):572–5. 

(6) Vallance P, Leone A, Calver A, Collier J, Moncada S. Endogenous Dimethylarginine as an Inhibitor of Nitric Oxide Synthesis. J Cardiovasc Pharmacol. 1992 Apr;20:S60–2. 

(7) Palmer RMJ, Ashton DS, Moncada S. Vascular endothelial cells synthesize nitric oxide from L-arginine. Nature. 1988 Jun;333(6174):664–6. 

We have now reworked and expanded this section by adding more citations (References 9,11 and 12) and referring to new studies (reference number 5, 9 and 10). The original version Lines 65-72- has now become revised version Lines 72-83 and states the following:

while the third isoform is inducible (iNOS). Asymmetric dimethylarginine (ADMA) is produced by all cells and is a product of protein degradation and competes with L-arginine for each of the three NOS isoforms (5). However, ADMA cannot be used as a substrate by NOS and as a result ADMA is a biologically significant inhibitor of NO production (6,7). 

As L-arginine is the precursor for NO synthesis (8) and ADMA a potent endogenous inhibitor of NO synthesis (6)(7), the arginine/ADMA plasma concentration ratio is widely considered to be an important indicator of NO bioavailability (9,10). In addition, L-arginine and ADMA have been shown to influence their human brain endothelial and umbilical vein endothelial cellular availability via transport interactions (11)(12). Together these facts imply that the interplay between the transport of these two cationic amino acids at the blood-brain barrier (BBB) and blood-cerebrospinal fluid (CSF) barriers is likely to directly relate to NO production at these interfaces and within the central nervous system (CNS). Interestingly,

(5) Wilcken DEL, Sim AS, Wang J, Wang XL. Asymmetric dimethylarginine (ADMA) in vascular, renal and hepatic disease and the regulatory role of l-arginine on its metabolism. Mol Genet Metab. 2007 Aug;91(4):309–17. 

(6) Vallance P, Leone A, Calver A, Collier J, Moncada S. Accumulation of an endogenous inhibitor of nitric oxide synthesis in chronic renal failure. Lancet. 1992 Mar 7;339(8793):572–5. 

(7) Vallance P, Leone A, Calver A, Collier J, Moncada S. Endogenous Dimethylarginine as an Inhibitor of Nitric Oxide Synthesis. J Cardiovasc Pharmacol. 1992 Apr;20:S60–2. 

(8) Palmer RMJ, Ashton DS, Moncada S. Vascular endothelial cells synthesize nitric oxide from L-arginine. Nature. 1988 Jun;333(6174):664–6.

(9) Bode-Boger, S.M.; Boger, R.H.; Kienke, S.; Junker, W.; Frolich, J.C. Elevated l-arginine/dimethylarginine ratio contributes to enhanced systemic NO production by dietary l-arginine in hypercholesterolemic rabbits. Biochem. Biophys. Res. Commun. 1996, 219, 598–603.

(10) Brinkmann SJ, Wörner EA, Buijs N, Richir M, Cynober L, van Leeuwen PA, Couderc R. The Arginine/ADMA Ratio Is Related to the Prevention of Atherosclerotic Plaques in Hypercholesterolemic Rabbits When Giving a Combined Therapy with Atorvastatine and Arginine. Int J Mol Sci. 2015 May 29;16(6):12230-42. doi: 10.3390/ijms160612230. PMID: 26035753; PMCID: PMC4490441.

(11) Shin S, Thapa SK, Fung HL. Cellular interactions between L-arginine and asymmetric dimethylarginine: Transport and metabolism. PLoS One. 2017 May 31;12(5):e0178710.

(12) Watson CP, Pazarentzos E, Fidanboylu M, Padilla B, Brown R, Thomas SA. The transporter and permeability interactions of asymmetric dimethylarginine (ADMA) and L-arginine with the human blood–brain barrier in vitro. Brain Res. 2016;1648:232–42. 

2. REVIEWER 1 comment

Original version Lines 127-147: Please provide references to support the advantages mentioned in the “in situ brain/choroid plexus perfusion” technique.

AUTHOR RESPONSE:

We have now added the following references to support each advantage as now stated in the 

revised version Lines 183-210.

These are:

Sanderson L, Khan A, Thomas S. Distribution of suramin, an antitrypanosomal drug, across the blood-brain and blood-cerebrospinal fluid interfaces in wild-type and P-glycoprotein transporter-deficient mice. Antimicrob Agents Chemother. 2007;51(9):3136–46. 

Thomas Née Williams SA, Segal MB. Identification of a saturable uptake system for deoxyribonucleosides at the blood-brain and blood-cerebrospinal fluid barriers. Brain Res. 1996;741(1–2).

3. REVIEWER 1 comment

- L163 (original version): How the CSF sample was taken is not clearly explained.

AUTHOR RESPONSE:

We have now explained the procedure in more detail. 

This can be found in the revised version Lines 226-229 and is as follows:

At the set perfusion time a CSF sample was taken by inserting a pulled glass micropipette connected to silicon tubing and a syringe into the cisterna magna. Gentle suction was applied, and CSF was withdrawn. Only clear CSF samples were processed and taken for analysis. Samples contaminated with blood were discarded. 

3. REVIEWER 1 comment

- Could the authors explain why they chose to perform one-tailed student-t tests instead of two-tailed and two-tailed test for one analysis (L490)? What are the significant differences based on two-tailed test, are they the same or different from current differences? Two-tailed test might be more reliable in terms of significance validity.

AUTHOR RESPONSE:

We have reviewed the statistical analysis performed on line 490 (original manuscript) for the comparison of [3H]arginine and [3H]ADMA and we can confirm that Student’s unpaired two tailed t-tests were performed to compare the two means at each time point (revised version Lines 605).

We chose not use a one tailed t-test as there was one lower value sample in the [3H]arginine group compared to the [3H]ADMA group samples at 10 minutes in most regions (except cerebellum).

4. REVIEWER 1 comment

 Discussion:

- L648-xxx: The authors mention that there might be a concentration gradient favouring flux of [3H]-arginine from the brain back into the plasma, however, the plasma concentrations of L-arginine is different in humans and mice. The adaptation of the results to humans could be different, how do the authors explain that [3H]-arginine sequestration to brain would be similar in humans in that case?

AUTHOR RESPONSE:

a)-As well as stating (revised version Lines 801-803) that the human and mouse plasma concentration of arginine is in the micromolar range (100 versus 140 �M, respectively) we now describe further on in the discussion (revised version L926 -L932) the kinetic constants for arginine transport in human and animal cells. They are similar in human and mouse cells. To do this we have also added references 52-54 listed below.

Revised version lines 926-932:

Furthermore, the human gene CAT-1 shares 87.6% amino acid identity with the mouse gene (52). Importantly, the high degree of sequence conservation has been shown to reflect functional similarity of the human transporters with their mouse homologues (53,54). L-arginine uptake by oocytes expressing hCAT-1 or mCAT-1 having a Km of 110-160 �M and 140-250 �M respectively (53,54)(55)(56). This would further suggest that there would be a functional similarity of the human and mouse CATs.

(52) Yoshimoto T, Yoshimoto E, Meruelo D. Molecular cloning and characterization of a novel human gene homologous to the murine ecotropic retroviral receptor. Virology. 1991 Nov;185(1):10–7.

(53) Closs EI, Lyons CR, Kelly C, Cunningham JM. Characterization of the third member of the MCAT family of cationic amino acid transporters. Identification of a domain that determines the transport properties of the MCAT proteins. Journal of Biological Chemistry. 1993 Oct;268(28):20796–800. 

(54) Closs EI, Gräf P, Habermeier A, Cunningham JM, Förstermann U. Human Cationic Amino Acid Transporters hCAT-1, hCAT-2A, and hCAT-2B: Three Related Carriers with Distinct Transport Properties ,. Biochemistry. 1997 May 1;36(21):6462–8. 

b) We have further clarified the following points (revised manuscript lines 916-926):

Interestingly, the cationic amino acids transport system, system-y+, has been detected at both luminal and abluminal membranes in brain capillary endothelial cells, with prevalence at the abluminal membrane (40). In addition, the half-saturation constants (Km) for arginine transport by murine CAT1 expressed in Xenopus oocytes and the y+-system expressed in human fibroblasts is 77 ± 2 μM and 40 ± 0.05 �M, respectively (32)(50)(49). It is therefore plausible that the transporters for arginine detected at the blood-CNS interfaces in our study are system-y+ transporters. We have previously identified the presence of the y+-system protein, CAT1, in human BBB cells (hCMEC/D3) using Western blotting and immunofluorescence (12)(51) and CAT 1 is expressed at the blood-CSF barrier specifically the rat choroid plexus (32)(41). 

5. REVIEWER 1 comment

- The changes of arginine concentrations might result in changes in ADMA concentrations endogenously in brain, rather than transport mechanisms, as they regulate one another’s metabolism. The opposite is also possible. What would be the authors’ comments about this? (Please see: https://doi.org/10.1016/j.ymgme.2007.04.017)

AUTHOR RESPONSE:

We gave now added the following information to the manuscript:

L1044 now states:

Although it must be noted that other mechanisms may also affect intracellular ADMA concentrations and/or NO production. For example, arginine inhibits ADMA catabolism by competing with ADMA for dimethylarginine dimethylaminohydrolase (DDAH) (67) and….

REVIEWER 2

Reviewer #2: The author has done tremendous research on the CNS delivery of L-arginine and ADMA. Overall, the manuscript sounds good. Please, see the mentioned below comments-

1. REVIEWER 2 comment: Mechanism of ADMA Transport: While the study provides valuable data on ADMA and L-arginine distribution, it would be strengthened by delving deeper into the specific mechanisms of ADMA transport across the blood-brain barrier. Are there specific transporters involved? Is there evidence of active transport or passive diffusion?

AUTHOR RESPONSE:

We have already explained that the self-inhibition studies indicated the presence of transporters for ADMA into the CNS. 

Amended version lines 938-943:

This suggests the use of transporters to transfer [3H]-ADMA across both mouse blood-brain and blood-CSF barriers. Importantly, this is the first study to reveal saturable transport of ADMA at the blood-CSF barrier (choroid plexus). It remains to be seen if these transporters are saturated by ADMA at physiological plasma concentrations which are 0.30-1.57 �M in humans (55) and 1.07-1.58 �M in mice (56). 

We have now also added the following information: 

Amended version Lines 826-839.

The concentration of [3H]-ADMA achieved in the CNS samples (except the choroid plexus, pineal and pituitary glands) was lower than that in the artificial plasma at all time points. This would indicate that transporters (and not just passive diffusion) were involved in the brain-to-plasma efflux of [3H]-ADMA. However, the CSF (choroid plexus)-to-blood transfer could involve transporters and/or just be passive diffusion of [3H]-ADMA. 

Amended version Lines 848-856.

This would suggest that multiple saturable transport mechanisms with different kinetics are involved in ADMA transfer across cell membranes and/or that the transporter is bidirectional and is capable of transferring ADMA into and out of the cell with differing affinities. Interestingly, system y+ is a bidirectional uniporter and can transfer positively charged amino acids with a Km for influx being lower than the Km for efflux(42). The results so far described would indicate that [3H]-ADMA can cross the cerebral capillary endothelium (i.e. the site of the BBB) and the blood-CSF barrier in both directions (i.e. blood-to-CNS and CNS-to-blood) and that transporters are involved, at least in the brain-to-blood transfer. 

Amended version Lines 1084-1087

In the future we hope to explore [3H]-arginine and [3H]-ADMA transport across the blood-brain and blood-CSF barriers in more detail determining the pharmacokinetic constants and the identity of the transporters involved. This may further contribute to understanding the contribution of ADMA efflux to any potential benefits of arginine supplementation. 

2. REVIEWER 2 comment: Specificity of Arginine Supplementation: The study suggests arginine supplementation as a potential therapeutic strategy. However, it would be beneficial to explore the specificity of this approach. Does arginine supplementation primarily increase L-arginine levels or also affect ADMA levels? Could this approach lead to unintended consequences?

AUTHOR RESPONSE:

a) Revised manuscript L1017 we now have added the following information (see red):

This present study would support the theory that arginine supplementation can increase NO production not by inhibiting arginine infl

---

## [Editor Report · Decision Letter 1]

9 Oct 2024

L -Arginine and asymmetric dimethylarginine (ADMA) transport across the mouse blood-brain and blood-CSF barriers: evidence of saturable transport at both interfaces and CNS to blood efflux.

PONE-D-24-21535R1

Dear Dr. Thomas,

We’re pleased to inform you that your manuscript has been judged scientifically suitable for publication and will be formally accepted for publication once it meets all outstanding technical requirements.

Kind regards,

Emre Avci

Academic Editor

PLOS ONE

---

## [Editor Report · Acceptance letter]

15 Oct 2024

PONE-D-24-21535R1 

PLOS ONE

Dear Dr. Thomas, 

I'm pleased to inform you that your manuscript has been deemed suitable for publication in PLOS ONE. Congratulations! Your manuscript is now being handed over to our production team.

Kind regards, 

on behalf of

Dr. Emre Avci 

Academic Editor

PLOS ONE